# Semi-Parametric Efficient Policy Learning with Continuous Actions

**Mert Demirer**
MIT
mdemirer@mit.edu

**Vasilis Syrgkanis**
Microsoft Research
vasy@microsoft.com

**Greg Lewis**
Microsoft Research
glewis@microsoft.com

**Victor Chernozhukov**
MIT
vchern@mit.edu

## Abstract

We consider off-policy evaluation and optimization with continuous action spaces. We focus on observational data where the data collection policy is unknown and needs to be estimated. We take a semi-parametric approach where the value function takes a known parametric form in the treatment, but we are agnostic on how it depends on the observed contexts. We propose a doubly robust off-policy estimate for this setting and show that off-policy optimization based on this estimate is robust to estimation errors of the policy function or the regression model. Our results also apply if the model does not satisfy our semi-parametric form, but rather we measure regret in terms of the best projection of the true value function to this functional space. Our work extends prior approaches of policy optimization from observational data that only considered discrete actions. We provide an experimental evaluation of our method in a synthetic data example motivated by optimal personalized pricing and costly resource allocation.

## 1 Introduction

We consider off-policy evaluation and optimization with continuous action spaces from observational data, where the data collection (logging) policy is unknown. We take a semi-parametric approach where we assume that the value function takes a known parametric form in the treatment, but we are agnostic on how it depends on the observed contexts/features. In particular, we assume that:

$$V(a, z) = \langle \theta_0(z), \phi(a, z) \rangle \tag{1}$$

for some known feature functions $\phi$ but unknown functions $\theta_0$. We assume that we are given a set of $n$ observational data points $(x_1, ..., x_n)$ that consist of i.i.d copies of the random vector $x = (y, a, z) \in \mathcal{Y} \times \mathcal{A} \times \mathcal{Z}$, such that $\mathbb{E}[y \mid a, z] = V(a, z)$.[1]

Our goal is to estimate a policy $\hat{\pi} : \mathcal{Z} \to \mathcal{A}$ from a space of policies $\Pi$ that achieves good regret:

$$\sup_{\pi \in \Pi} \mathbb{E}[V(\pi(z), z)] - \mathbb{E}[V(\hat{\pi}(z), z)] \leq R(\Pi, n) \tag{2}$$

for some regret rate that depends on the policy space $\Pi$ and the sample size $n$.

The semi-parametric value assumption allows us to formulate a doubly robust estimate $V_{DR}$ of the value function, from the observational data, which depends on first stage regression estimates of the coefficients $\theta_0(z)$ and the conditional covariance of the features $\Sigma_0(z) = \mathbb{E}[\phi(a,z)\phi(a,z)^T \mid z]$. The latter is the analogue of the propensity function when actions are discrete. Our estimate is doubly robust in that it is unbiased if either $\theta_0$ or $\Sigma_0$ is correct. Then we optimize this estimate:

$$\hat{\pi} = \sup_{\pi \in \Pi} V_{DR}(\pi) \tag{3}$$

**Main contributions.** We show that the double robustness property implies that our objective function satisfies a Neyman orthogonality criterion, which in turn implies that our regret rates depend only in a second order manner on the estimation errors on the first stage regression estimates of the functions $\theta_0, \Sigma_0$. Moreover, we prove a regret rate whose leading term depends on the variance of the difference of our estimated value between any two policy values within a "small regret-slice" and on the entropy integral of the policy space. We achieve this with a computationally efficient variant of the empirical risk minimization (ERM) algorithm (of independent interest) that uses a validation set to construct a preliminary policy and use it to regularize the policy computed on the training set. Based on this result, we manage to achieve variance-based regret bounds without the need for variance or moment penalization [15, 20, 9] used in prior work and which can render a computationally tractable policy learning problem, non-convex. Therefore, our method provides a computationally efficient alternative to the variance penalization when the original ERM problem is convex.[2] We also show that the asymptotic variance of our off-policy estimate (which governs the scale of the leading regret term) is asymptotic minimax optimal, in the sense that it achieves the semi-parametric efficiency lower bound.

**Robustness to mis-specification.** Notably, our approach provides meaningful guarantees even when our semi-parametric value function assumption is violated. Suppose that the true value function does not take the form of Equation (1), but rather takes some other form $V_0(a,z)$. Then one can consider the projection of this value function onto the forms of Equation (1), as:

$$\theta_p(z) = \arg\inf_{\theta} \mathbb{E}\left[(V_0(a,z) - \langle\theta(z),\phi(a,z)\rangle)^2 \mid z\right] \tag{4}$$

where the expectation is taken over the distribution of observed data. Then our approach takes the interpretation of achieving good regret bounds with respect to this best linear semi-parametric approximation. This is an alternative to the kernel smoothing approximation proposed by [20] in contextual bandit setting, as a regret target, and related to [12]. If there is some rough domain knowledge on the form of how the action affects the reward, then our semi-parametric approximate target should achieve better performance when the dimension of the action space is large, as the bias of kernel methods will typically incur an exponential in the dimension bias.

**Double robustness.** In cases where the collection policy is known, our doubly robust approach can be used for variance reduction via fitting first stage regression estimates to the policy value, whilst maintaining unbiasedness. Thus we can apply our approach to improve regret in the counterfactual risk minimization framework [20], [12] and as a variance reduction method in contextual bandit algorithms with continuous actions [20].

The problem we study in this paper is different in that we consider optimizing over continuous action spaces, rather than infinitesimal nudges, under semi-parametric functional form. This assumption is without loss of generality if treatment is binary or multi-valued. Hence, our results are a generalization of binary treatments to arbitrary continuous actions spaces, subject to our semi-parametric value assumption. In fact we show formally in the Appendix how one can recover the result of [1] for the binary setting, from our main regret bound.

**Related Literature.** Our work builds on the recent work at the intersection of semi-parametric inference and policy learning from observational data. The important work of [1] analyzes the binary treatments and infinitesimal nudges to continuous treatments. They also take a doubly robust approach so as to obtain regret bounds whose leading term depends on the semi-parametric efficient variance and the entropy integral and which is robust to first stage estimation errors. The problem we study in this paper is different in that we consider optimization over continuous action spaces, under a

semi-parametric functional form assumption on the payoff function. This paper is complementary to the work of [1], who allow for more general payoff functions, but restrict attention to the optimization of "infinitesimal nudges"[3] from the current actions. In the case of discrete actions the payoff function assumption is without loss of generality, and thus we are able to show formally in the Appendix that we can recover the result of [1] from our main regret bound. In turn our work builds on a long line of work on policy learning and counterfactual risk minimization [17, 26, 27, 1, 13, 28, 2, 7, 20, 12, 14]. Notably, the work of [28] extends the work of [1] to many discrete actions, but only proves a second moment based regret bound, which can be much larger than the variance. Our setting also subsumes the setting of many discrete actions and hence our regularized ERM offers an improvement over the rates in [28]. [9] formulates a general framework of statistical learning with a nuisance component. Our method falls into this framework and we build upon some of the results in [9]. However, for the case of policy learning the implications of [9] provide a variance based regret only when invoking second moment penalization, which can be intractable. We side-step this need and provide a computationally efficient alternative. Finally, most of the work on policy learning in machine learning assumes that the current policy (equiv. $\Sigma_0(z)$) is known. Hence, double robustness is used mostly as a variance reduction technique. Even for this literature, as we discuss above, our method can be seen an alternative of recent work on policy learning with continuous actions [12, 14] that makes use of non-parametric kernel methods.

Our work also connects to the semi-parametric estimation literature in econometrics and statistics. Our model is an extension of the partially linear model which has been extensively studied in the econometrics [8, 19]. By considering context-specific coefficients (random coefficients) and modeling a value function that is non-linear in treatment, we substantially extend the partially linear model. [24, 10] studied a special case of our model where output is linearly dependent on treatment given context, with the aim of estimating the average treatment effect. [10] constructed the doubly robust estimator and showed its semi-parametric efficiency under the linear-in-treatment assumption. We extend their results to a more general functional form and use the double-robustness property and semi-parametric efficiency for policy evaluation and optimization rather than treatment effect estimation. Our work is also connected to the recent and rapidly growing literature on the orthogonal/locally robust/debiased estimation literature [5, 6, 21].

## 2 Orthogonal Off-Policy Evaluation and Optimization

Let $\hat{\theta}$ be a first stage estimate of $\theta_0(z)$, which can be obtained by minimizing the square loss:

$$\hat{\theta} = \arg \inf_{\theta \in \Theta} \mathbb{E}_n \left[ (y - \langle \theta(z), \phi(a, z) \rangle)^2 \right] \tag{5}$$

where $\Theta$ is an appropriate parameter space for the parameters $\theta(z)$. Let $\Sigma_0(z)$ denote the conditional covariance matrix:

$$\Sigma_0(z) = \mathbb{E}[\phi(a, z) \, \phi(a, z)^T \mid z]$$

This is the analogue of the propensity model in discrete treatment settings. An estimate $\hat{\Sigma}(z)$ can be obtained by running a multi-task regression problem for each entry to the matrix, i.e.:

$$\hat{\Sigma}_{ij} = \arg \inf_{\Sigma_{ij} \in \mathcal{S}_{ij}} \mathbb{E} \left[ (\phi_i(a, z) \, \phi_j(a, z) - \Sigma_{ij}(z))^2 \right] \tag{6}$$

where $\mathcal{S}_{ij}$ is some appropriate hypothesis space for these regressions. Finally, the doubly robust estimate of the off-policy value takes the form:

$$V_{DR}(\pi) = \mathbb{E}_n \left[ v_{DR}(y, a, z; \pi) \right] \tag{7}$$

where:

$$v_{DR}(y, a, z; \pi) = \langle \theta_{DR}(y, a, z), \phi(\pi(z), z) \rangle \tag{8}$$

$$\theta_{DR}(y, a, z) = \hat{\theta}(z) + \hat{\Sigma}(z)^{-1} \phi(a, z) \, (y - \langle \hat{\theta}(z), \phi(a, z) \rangle) \tag{9}$$

The quantity $\theta_{DR}(y, a, z)$ can be viewed as an estimate of $\theta_0(z)$, based on a single observation. In fact, if the matrix $\hat{\Sigma}$ was equal to $\Sigma_0$, then one can see that $\theta_{DR}(y, a, z)$ is an unbiased estimate

of $\theta_0(z)$. Our estimate $v_{DR}$ also satisfies a doubly robust property, i.e. it is correct if either $\hat{\theta}$ is unbiased or $\hat{\Sigma}^{-1}$ is unbiased (see Appendix F for a formal statement). Finally, we will denote with $\theta_{DR}^0(y, a, z)$ the version of $\theta_{DR}$, where the nuisance quantities $\theta$ and $\Sigma$ are replaced by their true values, and correspondingly define $v_{DR}^0(y, a, z; \pi)$. We perform policy optimization based on this doubly robust estimate:

$$\hat{\pi} = \arg\max_{\pi \in \Pi} V_{DR}(\pi) \tag{10}$$

Moreover, we let $\pi_*^0$ be the optimal policy: $\pi_*^0 = \arg\max_{\pi \in \Pi} V(\pi)$.

**Remark 1** (Multi-Action Policy Learning). *A special case of our setup is the setting where the number of actions is finitely many. This can be encoded as $a \in \{e_1, \ldots, e_n\}$ and $\phi(a, z) = a$. In that case, observe that the covariance matrix becomes a diagonal matrix: $\Sigma_0(z) = \mathtt{diag}(p_1(z), \ldots, p_n(z))$, with $p_i(z) = \Pr[a = e_i \mid z]$. In this case, we simply recover the standard doubly robust estimate that combines the direct regression part with the inverse propensity weights part, i.e.:*

$$\theta_{DR,i}(y, a, z) = \hat{\theta}_i(z) + \frac{1}{\hat{p}_i(z)} \, 1\{a = e_i\} \, (y - \hat{\theta}_i(z))$$

*Thus our estimator is an extension of the doubly robust estimate from discrete to continuous actions.*

**Remark 2** (Finitely Many Possible Actions: Linear Contextual Bandits). *Another interesting special case of our approach is a generalization of the linear contextual bandit setting. In particular, suppose that there is only a finite (albeit potentially large) set of $N > p$ possible actions $A = \{a_1, \ldots, a_N\}$ and $a_i \in \mathbb{R}^p$. However, unlike the multi-action setting, where these actions are the orthonormal basis vectors, in this setting, each action $a_i \in A$, maps to a feature vector $\phi_i(z) := \phi(a_i, z)$. Then the reward $y$ that we observe satisfies $\mathbb{E}[y \mid z, a] = \langle \theta(z), \phi(a, z) \rangle$. This is a generalization of the linear contextual bandit setting, in which the coefficient vector $\theta(z)$ is a constant parameter $\theta$ as opposed to varying with $z$. In this case observe that: $\Sigma_0(z) = \sum_{i=1}^N p_i(z) \, \phi_i(z) \, \phi_i(z)^T = U D U^T$, i.e. it is the sum of $N$ rank one matrices where $D = \mathtt{diag}(p_1(z), \ldots, p_n(z))$, $p_i(z) = \Pr[a = a_i \mid z]$ and $U = [\phi_i(z), \ldots, \phi_N(z)]$ The doubly robust estimate of the parameter takes the form:*

$$\theta_{DR}(y, a, z) = \hat{\theta}(z) + (U D U^T)^{-1} \, \phi(a, z) \, (y - \langle \hat{\theta}(z), \phi(a, z) \rangle)$$

*This approach leverages the functional form assumption to get an estimate that avoids a large variance that depends on the number of actions $N$ but rather mostly depends on the number of parameters $p$. This is achieved by sharing reward information across actions.*

**Remark 3** (Linear-in-Treatment Value). *Consider the case where the value is linear in the action $\phi(a, z) = a \in \mathbb{R}^p$. In this case observe that: $\Sigma_0(z) = \mathrm{Var}(a \mid z)$. For instance, suppose that we assume that experimentation is independent across actions in the observed data. Then $\Sigma_0(z) = \mathtt{diag}(\sigma_1^2(z), \ldots, \sigma_p^2(z))$, where $\sigma_i^2 = \mathrm{Var}(a_i \mid z)$. Then the doubly robust estimate of the parameter takes the form:*

$$\theta_{DR,i}(y, a, z) = \hat{\theta}_i(z) + \frac{a_i}{\hat{\sigma}_i^2(z)} \, (y - \langle \hat{\theta}(z), a \rangle) \tag{11}$$

## 3 Theoretical Analysis

Our main regret bounds are derived for a slight variation of the ERM algorithm that we presented in the preliminary section. In particular, we crucially need to augment the ERM algorithm with a "validation" step, where we split our data into a training and validation step, and we restrict attention to policies that achieve small regret on the training data, while still maintaining small regret on the validation set. This extra modification enabled us to prove variance based regret bounds and

is reminiscent of standard approaches in machine learning, like $k$-fold cross-validation and early stopping, hence could be of independent interest.

---

**Algorithm 1:** Out-of-Sample Regularized ERM with Nuisance Estimates

---

1 The inputs are given by the sample of data $S = (x_1, \ldots, x_n)$, which we randomly split in two parts $S_1, S_2$. Moreover, we randomly split $S_2$ into validation and training samples $S_2^v$ and $S_2^t$.
2 Estimate the nuisance functions $\hat{\theta}(z)$ and $\hat{\Sigma}(z)$ using Equations (5) and (6) on $S_1$.
3 Use the output of Step 2 to construct the doubly robust moment in Equation (7) on $S_2^v$. Run ERM given in Equation (10) over policy space $\Pi_1$ on $S_2^v$ to learn a policy function $\pi_1$.
4 Use the output of Step 3 to construct a function class $\Pi_2$ defined as

$$\Pi_2 = \{\pi \in \Pi : \mathbb{E}_{S^v}[v_{DR}(y, a, z; \pi_1) - v_{DR}(y, a, z; \pi)] \leq \mu_n\}$$

for some $\mu_n$ and $\mathbb{E}_{S^v}$ denotes the empirical expectation over the validation sample.
5 Use the output of Step 1 to construct the doubly robust moment in Equation (7) on $S_2^t$. Run a constrained ERM on $S_2^t$ over $\Pi_2$.

---

We note that we present our theoretical results for the simpler case where the nuisance estimates are trained on a separate split of the data. However, our results qualitatively extend to the case where we use the cross-fitting idea of [5] (i.e. train a model on one half and predict on the other and vice versa).

**Regret bound.** To show the properties of this algorithm, we first show that the regret of the doubly robust algorithm is impacted in a second order manner by the errors in the first stage estimates. We will also make the following preliminary definitions. For any function $f$ we denote with $\|f\|_2 = \sqrt{\mathbb{E}[f(x)^2]}$, the standard $L^2$ norm and with $\|f\|_{2,n} = \sqrt{\mathbb{E}_n[f(x)^2]}$ its empirical analogue. Furthermore, we define the empirical entropy of a function class $H_2(\epsilon, \mathcal{F}, n)$ as the largest value, over the choice of $n$ samples, of the logarithm of the size of the smallest empirical $\epsilon$-cover of $\mathcal{F}$ on the samples with respect to the $\|\cdot\|_{2,n}$ norm. Finally, we consider the empirical entropy integral:

$$\kappa(r, \mathcal{F}) = \inf_{\alpha \geq 0} \left\{ 4\alpha + 10 \int_\alpha^r \sqrt{\frac{\mathcal{H}_2(\epsilon, \mathcal{F}, n)}{n}} d\epsilon \right\}, \tag{12}$$

Our statistical learning problem corresponds to learning over the function space:

$$\mathcal{F}_\Pi = \{v_{DR}(\cdot; \pi) : \pi \in \Pi\} \tag{13}$$

where the data is $x = (y, a, z)$. We will also make a very benign assumption on the entropy integral:

**ASSUMPTION 1.** *The function class $\mathcal{F}_\Pi$ satisfies that for any constant $r$, $\kappa(r, \mathcal{F}) \to 0$ as $n \to \infty$.*

**Theorem 1** (Variance-Based Oracle Policy Regret)**.** *Suppose that the nuisance estimates satisfy that their mean squared error is upper bounded w.p. $1 - \delta/2$ by $h_{n,\delta}^2$, i.e. w.p. $1 - \delta/2$ over the randomness of the nuisance sample:*

$$\max \left\{ \mathbb{E}[(\hat{\theta}(z) - \theta_0(z))^2], \mathbb{E}[\|\hat{\Sigma}(z) - \Sigma_0(z)\|_{Fro}^2] \right\} \leq h_{n,\delta}^2 \tag{14}$$

*Let $r = \sup_{\pi \in \Pi} \sqrt{\mathbb{E}[v_{DR}(z; \pi)^2]}$ and $\mu_n = \Theta\left(\kappa(r, \mathcal{F}_\Pi) + r\sqrt{\frac{\log(1/\delta)}{n}}\right)$. Moreover, let*

$$\Pi_*(\epsilon) = \{\pi \in \Pi : V(\pi_*^0) - V(\pi) \leq \epsilon\}, \tag{15}$$

*denote an $\epsilon$-regret slice of the policy space. Let $\epsilon_n = O(\mu_n + h_{n,\delta}^2)$ and*

$$V_2^0 = \sup_{\pi, \pi' \in \Pi_*(\epsilon_n)} Var(v_{DR}^0(x; \pi) - v_{DR}^0(x; \pi')) \tag{16}$$

*denote the variance of the difference between any two policies in an $\epsilon_n$-regret slice, evaluated at the true nuisance quantities. Then the policy $\pi_2$ returned by the out-of-sample regularized ERM, satisfies w.p. $1 - \delta$ over the randomness of $S$:*

$$V(\pi_*^0) - V(\pi_2) = O\left(\kappa(\sqrt{V_2^0}, \mathcal{F}_\Pi) + \sqrt{\frac{V_2^0 \log(1/\delta)}{n}} + h_{n,\delta}^2\right) \tag{17}$$

*Expected regret is $O\left(\kappa(\sqrt{V_2^0}, \mathcal{F}_\Pi) + \sqrt{\frac{V_2^0}{n}} + h_n^2\right)$, with $h_n^2$ is expected MSE of nuisance functions.*

We provide a proof of this Theorem in Appendix C. The regret result contains two main contributions: 1) first the impact of the nuisance estimation error is of second order (i.e. $h_{n,\delta}^2$ instead of $h_{n,\delta}$), 2) the leading regret term depends on the variance of small-regret policy differences and the entropy integral of the policy space. The first property stems from the Neyman orthogonality property of the doubly robust estimate of the policy. The second property stems from the out-of-sample regularization step that we added to the ERM algorithm. Typically, we will have $h_{n,\delta}^2 = o(1/\sqrt{n})$ and thereby this term is of lower order than the leading term. Moreover, for many policy spaces $\kappa(0, \mathcal{F}_\Pi) = 0$, in which case we see that if the setting satisfies a "margin" condition (i.e. the best policy is better by a constant $\Delta$ margin), then eventually the variance of small regret policies is $0$, since it only contains the best policy. In that case, our bound leads to fast rates of $\log(n)/n$ as opposed to $1/\sqrt{n}$, since the leading term vanishes (similar to the $\log(n)/n$ achieved in bandit problems with such a margin condition).

Dependence on the quantity $V_2^0$ is quite intuitive: if two policies have almost equivalent regret up to a $\mu_n$ rate, then it will be very easy to be mislead among them if one has much higher variance than the other. For some classes of problems, the above also implies a regret rate that only depends on the variance of the optimal policy (e.g. when all policies with low regret have a variance that is not much larger than the variance of the optimal policy. In Appendix G we show that the latter is always the case for the setting of binary treatment studied in [1] and therefore applying our main result, we recover exactly their bound for binary treatments.

**Semi-parametric efficient variance.** Our regret bound depends on the variance of our doubly robust estimate of the value function. One then wonders if there are other estimates of the value function that could achieve better variance than $V_{DR}(\pi)$. However, we show that at least asymptotically and without further assumptions on the functions $\theta_0(z)$ and $\Sigma_0(z)$, this cannot be the case. In particular, we show that our estimator achieves what is known as the semi-parametric efficient variance limit for our setting. More importantly for our regret result, this is also true for the semiparametric efficient variance of the policy differences. This is the case in our main setup; where the model is mis-specified and only a projection of the true value; and even if we assume that our model is correct, but make the extra assumption of homoskedasticity, i.e., the conditional variance of residuals of outcomes $y$ do not depend on $(a, z)$. Homoskedasticity is needed for efficiency because otherwise one can optimally re-weight the moments to obtain a more efficient estimator. Such optimally re-weighted estimators are typically avoided in practice as they heavily rely on the well-specification of the model.

**Theorem 2** (Semi-parametric Efficiency). *If the model is mis-specified, i.e. $V_0(a, z) \neq V(a, z)$ the asymptotic variance of $V_{DR}(\pi)$ is equal to the semi-parametric efficiency bound for the policy value $\langle \theta_p(z), \pi(z) \rangle$ defined in Equation (4). If the model is correctly specified, $V_{DR}(\pi)$ is semi-parametrically efficient under homoskedasticity, i.e. $\mathbb{E}[(y - V(a, z))^2 \mid a, z] = \mathbb{E}[(y - V(a, z))^2]$.*

We provide a proof for the value function, but this result also extends to the difference of values. We conclude the section by providing concrete examples of rates for policy classes of interest.

**Example 1** (VC Policies). *As a concrete example, consider the case when the class $\mathcal{F}_\Pi$ is a VC-subgraph class of VC dimension $d$ (e.g. the policy space has small VC-dimension or pseudo-dimension), and let $S = \mathbb{E}[\sup_\pi v_{DR}(x; \pi)^2]$. Then Theorem 2.6.7 of [22] shows that: $\mathcal{H}_2(\epsilon, \mathcal{F}_\Pi, n) = O(d(1 + \log(S/\epsilon)))$ (see also discussion in Appendix G). This implies that*

$$\kappa(r, \mathcal{F}_\Pi) = O\left(\int_0^r \sqrt{d(1 + \log(S/\epsilon))}d\epsilon\right) = O\left(r\sqrt{d}\sqrt{1 + \log(S/r)}\right).$$

*Hence, we can conclude that regret is $O\left(\sqrt{V_2^0(1 + \log(S/V_2^0))}\sqrt{\frac{d}{n}} + \sqrt{\frac{V_2^0 \log(1/\delta)}{n}} + h_{n,\delta}^2\right)$. For the case of binary action policies (as we discuss in Appendix G) this result recovers the result of [1] for binary treatments up to constants and extends it to arbitrary action spaces and VC-subgraph policies.*

**Example 2** (High-Dimensional Index Policies). *As an example, we consider the class of policies, characterized by a constant number of $\ell_1$ or $\ell_0$-bounded linear indices:*

$$\Pi_1 = \{z \to \Gamma(\langle \beta_1, z \rangle, \ldots, \langle \beta_d, z \rangle) : \beta_i \in \mathbb{R}^p, \|\beta_i\|_1 \leq s\} \tag{18}$$

*where $\Gamma : \mathbb{R}^d \to \mathbb{R}^m$ is a fixed L-Lipschitz function of the indices, with $d, m$ constants, while $p >> n$ (and similarly for $\Pi_0$, where use $\|\beta_i\|_0 \leq s$). Assuming $v_{DR}(y, a, z; \pi)$ is a Lipschitz function of $\pi(z)$ and since $\Gamma$ is a Lipschitz function of $\langle \beta, z \rangle$, we have by a standard multi-variate Lipschitz*

*contraction argument (and since $d$, $m$ are constants), that the entropy of $\mathcal{F}_\Pi$ is of the same order as the maximum entropy of each of the linear index spaces: $B_1 := \{z \to \langle \beta_i, z \rangle : \beta \in \mathbb{R}^p, \|\beta_i\|_1 \le s\}$. Moreover, by known covering arguments (see e.g. [25], Theorem 3) that if $\|z\|_\infty \le 1$, then:*

$$\mathcal{H}_2(\epsilon, B, n) = O\left(\frac{s^2 \log(d)}{\epsilon^2}\right). \quad \textit{Thus we get } \kappa(r, \mathcal{F}_\Pi) = O\left(s \log(n) \sqrt{\frac{\log(d)}{n}} + \frac{r}{n}\right), \textit{ which}$$

*leads to regret* $O\left(s \log(n) \sqrt{\frac{\log(d)}{n}} + \sqrt{\frac{V_2^0 \log(1/\delta)}{n}} + h_{n,\delta}^2\right)$. *In this setting, we observe that the policy space is too large for the variance to drive the asymptotic regret. There is a leading term that remains even if the worse-case variance of policies in a small-regret slice is $0$. Intuitively this stems from the high-dimensionality of the linear indices, which introduces an extra dimension of error, namely bias due to regularization. On the contrary, for exactly sparse policies $B_0 := \{z \to \langle \beta_i, z \rangle : \beta \in \mathbb{R}^p, \|\beta_i\|_0 \le s\}$, we have that since for any possible support the entropy at scale $\epsilon$ is at most $O(s \log(1/\epsilon))$, we can take a union over all $\binom{p}{s} \le \left(\frac{ep}{s}\right)^s$ possible sparse supports, which implies $\mathcal{H}_2(\epsilon, \mathcal{F}_\Pi, n) = O(s(\log(d/s) + \log(1/\epsilon))$.*

*Thus $\kappa(r, \mathcal{F}_\Pi) = O\left(r\sqrt{\log(1/r)}\sqrt{\frac{s\log(d/s)}{n}}\right)$, leading to policy regret similar to the VC classes:*

$$O\left(\sqrt{V_2^0 \log(1/V_2^0)}\sqrt{\frac{s\log(d/s)}{n}} + \sqrt{\frac{V_2^0 \log(1/\delta)}{n}} + h_{n,\delta}^2\right).$$

**Remark 4** (Instrumental Variable Estimation)**.** *Our main regret results extend to the instrumental variables settings where treatments are endogenous but we have a vector of instrumental variables $w$ satisfying*

$$\mathbb{E}[(y - \langle \theta_0(z), \phi(a,z) \rangle)w \mid z] = 0,$$

*and $\Sigma_0^I(z) = \mathbb{E}[w\phi(a,z)^T \mid z]$ is invertible. Then we can use the following doubly robust moment*

$$\theta_{DR,I}(y,a,z,w) = \hat{\theta}(z) + \hat{\Sigma}^I(z)^{-1} w(y - \langle \hat{\theta}(z), \phi(a,z) \rangle).$$

**Remark 5** (Estimating the First Stages)**.** *Bounds on first stage errors as a function of sample complexity measures can be obtained by standard results on the MSE achievable by regression problems (see e.g. [18, 23]). Essentially these are bounds for the regression estimates $\hat{\theta}(z)$ and $\hat{\Sigma}(z)$, as a function of the complexity of their assumed hypothesis spaces. Since the latter is a standard statistical learning problem that is orthogonal to our main contribution, we omit technical details.[4] Since the square loss is a strongly convex objective the rates achievable for these problems are typically fast rates on the MSE (e.g. $h_{n,\delta}^2$ is of the order $1/n$ for the case of parametric hypothesis spaces, and typically $o(1/\sqrt{n})$ for reproducing kernel Hilbert spaces with fast eigendecay (see e.g. [23])). Thus the term $h_{n,\delta}^2$ is of lower order. For instance, the required rates for the term $h_{n,\delta}^2$ to be of second order in our regret bounds are achievable if these nuisance regressions are $\ell_1$-penalized linear regressions and several regularity assumptions are satisfied by the data distribution, even when the dimension $p$ of $z$ is growing with $n$.*

**Extension: Semi-Bandit Feedback** Suppose that our value function takes the form: $V(a,z) = \phi(a,z)^T \Theta_0(z) \phi(a,z)$, where $\Theta_0(z)$ is a $p \times p$ matrix and we observe semi-bandit feedback, i.e. we observe a vector $Y$ s.t.: $\mathbb{E}[Y \mid a,z] = \Theta_0(z)^T \phi(a,z)$. Then we can apply our DR approach to each coordinate of $Y$ separately.

$$V_{DR}(\pi) = \mathbb{E}_n\left[\phi(\pi(z),z)^T \left(\hat{\Theta}(z) + \hat{\Sigma}(z)^{-1} \phi(a,z)(Y^T - \phi(a,z)^T\hat{\Theta}(z))\right) \phi(\pi(z),z)\right]$$

All the theorems in this section extend to this case, which will prove useful in our pricing application where $a$ is the price of a set of products and $Y$ is the vector of observed demands for each product.

## 4 Application: Personalized Pricing

Consider the personalized pricing of a single product. The objective is the revenue:

$$V(p,z) = p(a(z) - b(z)p)$$

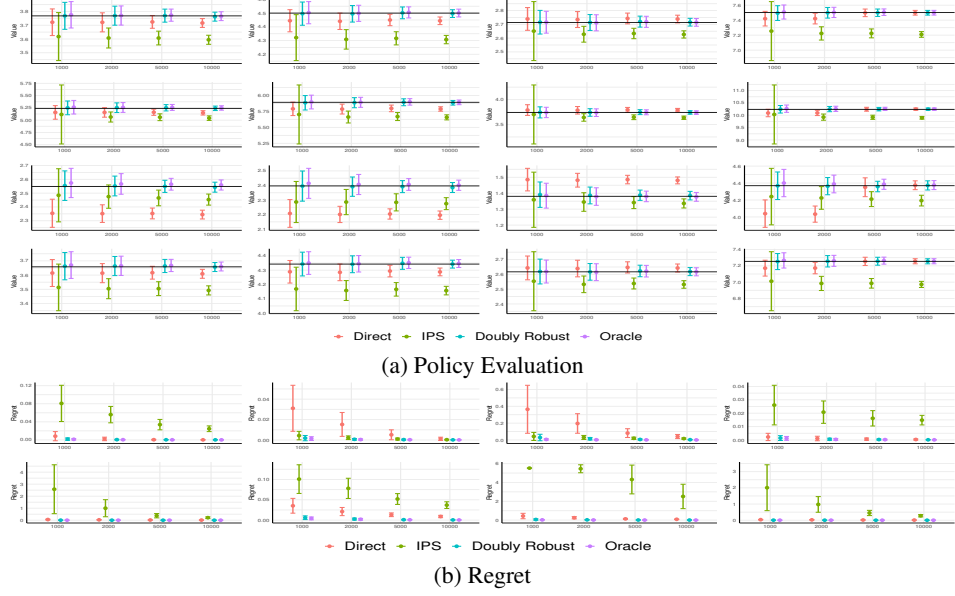

(a) Policy Evaluation

(b) Regret

Figure 1: (a) Black line shows the true value of the policy and each line shows the mean and standard deviation of the value of the corresponding policy over 100 simulations. (b) Each line shows the mean and standard deviation of regret over 100 simulations. The top half reports the regret for a constant policy, the bottom half reports regret for a linear policy.

where $b(z) \geq \gamma > 0$ and $a(z) + b(z)p$ gives the unknown, context-specific demand function. We assume that we observe an unbiased estimate $d$ of demand:

$$\mathbb{E}[d \mid z, p] = a(z) - b(z)\,p$$

We want to optimize over a space of personalized pricing policies $\Pi$. If, for instance, the observational policy was homoskedastic (i.e. the exploration component was independent of the context $z$), we show in Appendix H that doubly robust estimators for $a(z)$ and $b(z)$ are

$$a_{DR}(z) = \hat{a}(z) + \left(1 + \hat{g}(z)\frac{\hat{g}(z) - p}{\hat{\sigma}^2}\right)(d - \hat{a}(z) - \hat{b}(z)\,p)$$

$$b_{DR}(z) = \hat{b}(z) + \frac{p - \hat{g}(z)}{\hat{\sigma}^2}(d - \hat{a}(z) - \hat{b}(z)\,p)$$

where $g(z) = \mathbb{E}[p \mid z]$ and the variance $\sigma^2$. Thus, in this example, we only need to estimate the mean treatment policy $\mathbb{E}[p \mid z]$ and the variance $\sigma^2$.

**Experimental evaluation.** We empirically evaluate our framework on the personalized pricing application with synthetic data. In particular, we use simulations to assess our estimator's ability to evaluate and optimize personalized pricing functions. To do this, we compare the performance of our doubly robust estimator with (1) Direct estimator, $\langle \hat{\theta}(z), \phi(a, z) \rangle$, (2) Inverse propensity score estimator [5], (3) Oracle orthogonal estimator, $v_{DR}^o(x, \pi)$.

**Data Generating Process.** Our simulation design considers a sparse model. We assume that there are $k$ continuous context variables distributed uniformly $z_i \sim U(1, 2)$ for $i = 1, \ldots, k$ but only $l$ of them affects demand. Let $\bar{z} = 1/l(z_i + \cdots + z_l)$. Price $p$ and demand $d$ are generated as $x \sim \mathcal{N}(\bar{z}, 1), d = a(\bar{z}) - b(\bar{z})x + \epsilon$ and $\epsilon \sim \mathcal{N}(0, 1)$. We consider four functional forms for the demand model: (i) (Quadratic) $a(z) = 2z^2, b(z) = 0.6z$, (ii) (Step) $a(z) = 5\{z < 1.5\} + 6\{z > 1.5\}, b(z) = 0.7\{z < 1.5\} + 1.2\{z > 1.5\}$, (iii) (Sigmoid) $a(z) = 1/(1 + \exp(z)) + 3, b(z) = 2/(1 + \exp(z)) + 0.1$, (iv) (Linear) $a(z) = 6z, b(z) = z$

These functions and the data generating process ensure that the conditional expectation function of demand given $z$ is non-negative for all $z$, the observed prices are positive with high probability,

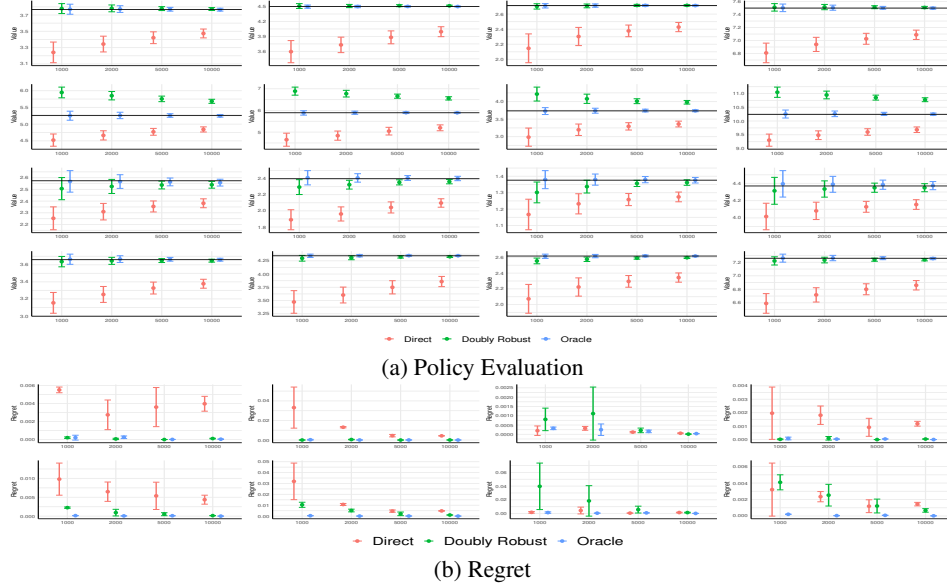

(a) Policy Evaluation

(b) Regret

Figure 2: Quadratic, Low Dimensional Regime: (a) Black line shows the true value of the policy, each line shows the mean and standard deviation of the value of the corresponding policy over 100 simulations. (b) Mean and standard deviation of regret reported over 100 simulations. We omit the results for the inverse propensity score method since they are too large to report together with the other estimates in the high dimensional regime.

and the optimal prices are in the support of the observed prices. In each experiment, we generate 1000, 2000, 5000, and 10000 data points, and report results over 100 simulations. We estimate the nuisance functions using 5-fold cross-validated lasso model with polynomials of degrees up to 3 and all the two-way interactions of context variables. We present the results for two regimes: (i) Low dimensional with $k = 2, l = 1$, (ii) High dimensional with $k = 10, l = 3$.

**Policy Evaluation.** For policy evaluation we consider four pricing functions: (i) Constant, $\pi(z) = 1$, (ii) Linear, $\pi(z) = z$, (iii) Threshold, $\pi(z) = 1 + 1\{z > 1.5\}$, (iv) Sin, $\pi(z) = \sin(z)$. The results for the low dimensional regime are summarized in Figure 1(a), where each row and column corresponds to a different demand function and a policy function, respectively[6]. The results show that, as expected, our the performance of our method is very similar to the oracle estimator and achieves a significantly better performance than the direct and inverse propensity score methods, which suffer from large biases. These results also support our claim that the asymptotic variance of the doubly robust estimate is the same as the variance of the oracle method. It is also important to point out that we obtain similar performances across two different regimes.

**Regret.** To investigate the regret performance of our method, we consider a constant pricing function, $\pi(z) = \gamma$ and a linear policy $\pi(z) = (\gamma_1 z_1 + \cdots + \gamma_k z_k)$. We compute the optimal pricing functions in these function spaces and report the distribution of regret in Figure 4(b) under the low dimensional regime and in Appendix I under the high dimensional regime. Across the four demand functions and two pricing functions, our method achieves small regrets, comparable to the oracle. The direct and inverse propensity methods, depending on the demand function, yield large regrets.

**Quadratic Model** Finally, we consider the same simulation exercise under the assumption that an unbiased estimate of revenue rather than demand is observed. Since revenue depends on the $p^2$ the model is now quadratic $r = a(z)p - b(z)p^2 + \epsilon$. For the data generating process we use the same functions $a(z)$ and $b(z)$ as in the personalized pricing example [7]. Figures 2 and 5 in Appendix I summarize results for policy evaluation and optimization. The overall performance of our doubly robust estimator is similar to the demand model, and it performs better the direct model.

## Footnotes

[1]In most of the paper, we can allow for the case where $z$ is endogenous, in the sense that $\mathbb{E}[y \mid a, z] = V(a, z) + f_0(z)$. In other words, the noise in the random variable $y$ can be potentially correlated with $z$. However, we assume that conditional on $z$, there is no remaining endogeneity in the choice of the action in our data. The latter is typically referred to as conditional ignorability/exogeneity [11].

[2]We provide two examples of convex problem in our experiments.

[3]Infinitesimal nudges are defined as $\epsilon$ positive or negative perturbations over a baseline policy. With the limit of epsilon going to zero the latter is essentially a binary policy problem where the binary choice is whether to $\epsilon$ increase or decrease the current treatment level.

[4] Also, as we show in pricing application in some problems simpler estimators arise for co-variance matrix.

[5] $\theta_{IPS}(y, a, z) = \hat{\Sigma}(z)^{-1}\,\phi(a, z)\,y$

[6]The results are very similar for the high dimensional model which are reported in Figure 4(a) in the appendix.

[7]We provide the calculation of the doubly robust estimators for this example in Appendix H.

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
