[Supplementary Material]

## A  Proof of Universal Orthogonality Lemma

We first start by defining a sufficient condition for the notion of *universal orthogonality* of a loss function, as defined by [9]. A loss function $L(\pi; h) = \mathbb{E}[\ell(x, \pi(z); h(z))]$ is universally orthogonal with respect to $h$ if for any $\pi \in \Pi$:

$$\mathbb{E}[\nabla_{h(z),\pi(z)}\ell(x, \pi(z); h_0(z)) \mid z] = 0 \tag{20}$$

where $h_0$ is the true value of the nuisance parameter $h$.

**Lemma 3.** *The loss function* $L(\pi; h) = -\mathbb{E}[\langle \theta_{DR}(y, a, z), \phi(\pi(z), z)\rangle]$ *is universally orthogonal with respect to* $h = (\theta, \Sigma)$.

*Proof.* We show that the population loss function that corresponds to the doubly robust estimate, satisfies the universal orthogonality condition. For simplicity of notation let $K(z) = \Sigma(z)^{-1}$. Then the population loss is:

$$V_{DR}^0(\pi; \hat{\theta}, \Sigma^{-1}) = \mathbb{E}\left[\left\langle \hat{\theta}(z) + \Sigma^{-1}(z)\,\phi(a, z)\,(y - \langle\hat{\theta}(z), \phi(a, z)\rangle), \phi(\pi(z), z)\right\rangle\right]$$

Let:

$$\beta(a, z, \xi, K) = \xi + K\phi(a, z)\,(y - \langle\xi, \phi(a, z)\rangle)$$

Observe that:

$$V_{DR}^0(\pi; \hat{\theta}, \Sigma^{-1}) = \mathbb{E}\left[\left\langle \beta(a, \hat{\theta}(z), \Sigma^{-1}(z)), \phi(\pi(z), z)\right\rangle\right]$$

To show universal orthogonality it suffices to show that:

$$\mathbb{E}\left[\nabla_{\xi, K}\beta(a, z, \theta_0(z), \Sigma_0^{-1}(z)) \mid z\right] = 0$$

This follows easily by simple algebraic manipulations:

$$\begin{aligned}
\mathbb{E}\left[\nabla_\xi\beta(a, z, \theta_0(z), \Sigma_0^{-1}(z)) \mid z\right] &= \mathbb{E}\left[\mathbb{I} - \Sigma_0^{-1}(z)\,\phi(a, z)\phi(a, z)^T \mid z\right] \\
&= \mathbb{I} - \Sigma_0^{-1}(z)\mathbb{E}\left[\phi(a, z)\phi(a, z)^T \mid z\right] = \mathbb{I} - \Sigma_0^{-1}(z)\,\Sigma_0(z) = 0
\end{aligned}$$

and

$$\mathbb{E}\left[\nabla_{K_{ij}}\beta(a, \theta_0(z), \Sigma_0^{-1}(z)) \mid z\right] = \mathbb{E}\left[\phi_j(a, z)\,(y - \langle\theta_0(z), \phi(a, z)\rangle) \mid z\right]$$

Now observe that since $\theta_0(z)$ is the minimizer of the conditional squared loss, taking the first order condition implies:

$$\begin{aligned}
\mathbb{E}[(V_0(a, z) - \langle\theta_0(z), \phi(a, z)\rangle)\,\phi(a, z) \mid z] = 0 &\iff \\
\mathbb{E}[V_0(a, z)\,\phi(a, z) \mid z] &= \mathbb{E}[\langle\theta_0(z), \phi(a, z)\rangle)\,\phi(a, z) \mid z]
\end{aligned}$$

Moreover:

$$\mathbb{E}[y\,\phi(a, z) \mid z] = \mathbb{E}[\mathbb{E}[y \mid a, z]\,\phi(a, z)] = \mathbb{E}[V_0(a, z)\,\phi(a, z)]$$

Combining the two yields:

$$\mathbb{E}\left[\phi(a, z)\,(y - \langle\theta_0(z), \phi(a, z)\rangle) \mid z\right] = 0$$

which implies orthogonality with respect to $K$. □

## B  Proof of Main Regret Theorem 1

We first consider an arbitrary empirical loss minimization problem of the form:

$$f_n = \arg\min_{f \in \mathcal{F}} \mathbb{E}_n[f(x)] := \frac{1}{n}\sum_{i=1}^n f(x_i) \tag{21}$$

where $x_i \in \mathcal{X}$ are i.i.d. drawn from an unknown distribution and $\mathcal{X}$ is an arbitrary data space. Throughout the section we will assume that: $\sup_{f \in \mathcal{F}}|f(x)| \leq 1$. All the results can be generalized

to the case of $\sup_{f \in \mathcal{F}} |f(x)| \le R$, for some arbitrary $R$, by simply first re-scaling the losses, and then invoking the theorems of this section.

We will also make the following preliminary definitions. For any function $f$ we denote with $\|f\|_2 = \sqrt{\mathbb{E}[f(x)^2]}$, the standard $L^2$ norm and with $\|f\|_{2,n} = \sqrt{\mathbb{E}_n[f(x)^2]}$ its empirical analogue. The localized Rademacher complexity is the defined as:

$$\mathcal{R}(r, \mathcal{F}) = \mathbb{E}_{\epsilon, x_{1:n}} \left[ \sup_{f \in \mathcal{F}: \|f\|_2 \le r} \frac{1}{n} \sum_{i=1}^n \epsilon_i \, f(x_i) \right] \tag{22}$$

where $\epsilon_i$ are independent Rademacher variables that take values $\{-1, 1\}$ with equal probability.

Furthermore, we define the empirical entropy of a function class $H_2(\epsilon, \mathcal{F}, n)$ as the largest value, over the choice of $n$ samples, of the logarithm of the size of the smallest empirical $\epsilon$-cover of $\mathcal{F}$ on the samples with respect to the $\|\cdot\|_{2,n}$ norm. Finally, we consider the empirical entropy integral defined as:

$$\kappa(r, \mathcal{F}) = \inf_{\alpha \ge 0} \left\{ 4\alpha + 10 \int_\alpha^r \sqrt{\frac{\mathcal{H}_2(\epsilon, \mathcal{F}, n))}{n}} d\epsilon \right\}, \tag{23}$$

Throughout this section we will make the following benign assumption that essentially makes the problem *learnable*:

**ASSUMPTION 1.** *The function class satisfies that for any constant $r$, $\kappa(r, \mathcal{F}) \to 0$ as $n \to \infty$*

We will use the following theorems from the prior work of [9] as a starting point as they are formalized in manner convenient for our problem.

**Theorem 4** (Foster, Syrgkanis [9], Theorem 4). *Consider any function class $\mathcal{F} : \mathcal{X} \to [-1, 1]$ and let $f_n$ be the outcome of the constrained ERM. Pick any $f_* \in \mathcal{F}$ and let $r = \sup_{f \in \mathcal{F}} \|f - f_*\|_2$. Then for some constants $C_1, C_2$ and for any $\delta > 0$, w.p. $1 - \delta$:*

$$\mathbb{E}[f_n(x) - f_*(x)] \le C_1 \left( \mathcal{R}(r, \mathcal{F} - f^*) + r\sqrt{\frac{\log(1/\delta)}{n}} + \frac{\log(1/\delta)}{n} \right)$$

$$\le C_1 C_2 \left( \kappa(r, \mathcal{F}) + r\sqrt{\frac{\log(1/\delta)}{n}} + \frac{\mathcal{H}_2(r, \mathcal{F}, n)}{n} + \frac{\log(1/\delta)}{n} \right).$$

**Lemma 5** (Foster, Syrgkanis [9], Lemma 4). *Consider a function class $\mathcal{F} : \mathcal{X} \to [-1, 1]$ and pick any $f_* : \mathcal{X} \to [-1, 1]$ (not necessarily in $\mathcal{F}$). Moreover, let:*

$$Z_n(r) = \sup_{f \in \mathcal{F}: \|f - f^*\|_2 \le r} |\mathbb{E}_n[f(x) - f_*(x)] - \mathbb{E}[f(x) - f_*(x)]| \tag{24}$$

*Then for some constant $C_3$ and for any $\delta > 0$, w.p. $1 - \delta$:*

$$Z_n(r) \le C_3 \left( \mathcal{R}(r, \mathcal{F} - f^*) + r\sqrt{\frac{\log(1/\delta)}{n}} + \frac{\log(1/\delta)}{n} \right)$$

Our goal is to replace $r$ in the latter Theorem with the worst-case variance of the functions $f \in \mathcal{F}$ in a small "regret"-ball around the optimal. We will achieve this by considering a slight modification of the ERM algorithm. In particular, we will split the data in half, and we will use one half as a *regularization sample* and the other half as the *training sample*. In particular, we will find the optimal function on the training sample, within the class of functions that also have relatively small regret on the regularization sample.

**Out-of-Sample Regularized ERM**  Consider the following algorithm:

- We split the samples $S$ in two parts $S_1$, $S_2$ and let $\mathbb{E}_{n_1}[\cdot]$ and $\mathbb{E}_{n_2}[\cdot]$ denote the corresponding empirical expectations.

- We run ERM over $\mathcal{F}$ on the first half and let $f_1$ be the outcome.

- Then we define the class of functions that have the constraint that they don't achieve much worse value than $f_1$ on the first half, i.e. we regularize policies based on their regret on the first half. More formally, for some constant $\mu_n$ to be defined later:

$$\mathcal{F}_2 = \{ f \in \mathcal{F} : \mathbb{E}_{n_1}[f(x) - f_1(x)] \le \mu_n \} \tag{25}$$

417 • Then we run constrained ERM on the second sample over the function space $\mathcal{F}_2$:

$$f_2 = \arg\min_{f \in \mathcal{F}_2} \mathbb{E}_{n_2}[f(x)] \tag{26}$$

418 **Theorem 6** (Variance-Based Regret). *Let $f_* = \arg\min_{f \in \mathcal{F}} \mathbb{E}[f(x)]$, $r = \sup_{f \in \mathcal{F}} \|f\|_2$ and choose*

419 $\mu_n = C \left( \kappa(r, \mathcal{F}) + r\sqrt{\frac{\log(6/\delta)}{n}} + \frac{\mathcal{H}_2(r, \mathcal{F}, n)}{n} + \frac{\log(6/\delta)}{n} \right)$, *with $C = 8 \max\{C_1 C_2, C_3 C_2\}$. Then,*

420 *w.p. $1 - \delta$ over the sample $S$, the outcome $f_2$ of the Out-of-Sample Regularized ERM satisfies:*

$$\mathbb{E}[f_2(x) - f_*(x)] = O\left( \kappa(\sqrt{V_2}, \mathcal{F}_*(\mu_n)) + \sqrt{\frac{V_2 \log(3/\delta)}{n}} \right) \tag{27}$$

421 *with: $\mathcal{F}_*(\mu_n) = \{f \in \mathcal{F} : \mathbb{E}[f(x) - f_*(x)] \leq \mu_n\}$ and $V_2 = \sup_{f \in \mathcal{F}_*(\mu_n)} Var(f(x) -$*
422 *$f_*(x))$. Moreover, the expected regret, in expectation over the samples $S_1, S_2$ is also of order*

423 $O\left( \kappa(\sqrt{V_2}, \mathcal{F}) + \sqrt{\frac{V_2}{n}} \right)$.

424 *Proof.* First we argue that w.p. $1 - \delta/6$, $f_* \in \mathcal{F}_2$. By the choice of $\mu_n$ and Theorem 4, we know that
425 w.p. $1 - \delta/4$ over the randomness of sample $S_1$:

$$\mathbb{E}[f_1(x) - f_*(x)] \leq \mu_n/2 \tag{28}$$

426 Moreover, by Lemma 5, w.p. $1 - \delta/6$ over the randomness of sample $S_1$:

$$\sup_{f \in \mathcal{F}} |\mathbb{E}_{n_1}[f(x) - f_*(x)] - \mathbb{E}[f(x) - f_*(x)]| \leq \mu_n/2$$

427 Combining the latter two properties we have, w.p. $1 - \delta/3$:

$$|\mathbb{E}_{n_1}[f_*(x) - f_1(x)]| \leq |\mathbb{E}[f_*(x) - f_1(x)]| + \mu_n/2 \leq \mu_n$$

428 Thus in this event, $f_* \in \mathcal{F}_2$.

429 Applying Theorem 4 for the last stage of the algorithm with function space $\mathcal{F}_2$ and conditioning on
430 the event that the first stage sample is such that $f_* \in \mathcal{F}_2$, we have that with probability $1 - \delta/3$ over
431 the randomness of the second sample:

$$\mathbb{E}[f_2(x) - f_*(x)] = C_1 C_2 \left( \kappa(r_2, \mathcal{F}_2) + r_2\sqrt{\frac{\log(3/\delta)}{n}} + \frac{\mathcal{H}_2(r_2, \mathcal{F}_2, n)}{n} + \frac{\log(3/\delta)}{n} \right)$$

432 where $r_2 = \sup_{f \in \mathcal{F}_2} \|f\|_2$. Thus by a union bound we get that with probability $1 - 2\delta/3$ over the
433 randomness of both samples, the latter bound holds.

434 Observe that for $f \in \mathcal{F}_2$, by Lemma 5, w.p. $1 - \delta/6$ over the first sample:

$$\sup_{f \in \mathcal{F}} |\mathbb{E}_{n_1}[f(x) - f_1(x)] - \mathbb{E}[f(x) - f_1(x)]| \leq 2 \sup_{f \in \mathcal{F}} |\mathbb{E}_{n_1}[f(x)] - \mathbb{E}[f(x)]| \leq \mu_n/2$$

435 Thus w.p. $1 - \delta/6$, $\mathcal{F}_2$ is a subset of the class:

$$\mathcal{F}_2^0 = \{f \in \mathcal{F} : |\mathbb{E}[f(x) - f_1(x)]| \leq \mu_n/2\} \tag{29}$$

436 Moreover, since $f_1$ has small regret, we know by the triangle inequality, for all $f \in \mathcal{F}_2^0$, w.p. $1 - \delta/3$:

$$|\mathbb{E}[f(x) - f_*(x)]| \leq |\mathbb{E}[f(x) - f_1(x)]| + |\mathbb{E}[f_1(x) - f_*(x)]| \leq \mu_n$$

437 Thus w.p. $1 - \delta/3$, $\mathcal{F}_2^0$ is in turn a subset of the function space:

$$\mathcal{F}_*(\mu_n) = \{f \in \mathcal{F} : |\mathbb{E}[f(x) - f_*(x)]| \leq \mu_n\}$$

438 which is a space of policies with regret at most $\mu_n$.

439 Thus we have that w.p. $1 - \delta/3$ over the first sample:

$$\begin{aligned}
r_2^2 &= \sup_{f \in \mathcal{F}_2} \mathbb{E}[(f(x) - f_*(x))^2] \leq \sup_{f \in \mathcal{F}_*(\mu_n)} \mathbb{E}[(f(x) - f_*(x))^2] \\
&= \sup_{f \in \mathcal{F}_*(\mu_n)} \left( Var(f(x) - f_*(x)) + \mathbb{E}[f(x) - f_*(x)]^2 \right) \\
&\leq \sup_{f \in \mathcal{F}_*(\mu_n)} Var(f(x) - f_*(x)) + \mu_n^2
\end{aligned}$$

440    We thus have that:

$$r_2 = \sqrt{\sup_{f \in \mathcal{F}_*(\mu_n)} \mathrm{Var}(f(x) - f_*(x))} + 2\mu_n = \sqrt{V_2} + 2\mu_n \qquad (30)$$

441    Combining the latter with the regret bound for $f_2$ (excluding lower order terms in $n$) we have that
442    w.p. $1 - \delta$:

$$\mathbb{E}[f_2(x) - f_*(x)] = O\left(\kappa(\sqrt{V_2} + 2\mu_n, \mathcal{F}_*(\mu_n)) + \sqrt{\frac{V_2 \log(3/\delta)}{n}}\right)$$

443    Moreover, using the concavity of the entropy integral with respect to its first argument, we have that:

$$\kappa(\sqrt{V_2} + 2\mu_n, \mathcal{F}) \leq \kappa(\sqrt{V_2}, \mathcal{F}_*(\mu_n)) + 2\mu_n \sqrt{\frac{H_2(\sqrt{V_2}, \mathcal{F}, n)}{n}} \qquad (31)$$

444    Since $\kappa(r, \mathcal{F}) \to 0$, we have that $\mu_n = o(1)$ and $H_2(\sqrt{V_2}, \mathcal{F}, n)$ is a constant. Thus, the second term
445    decays faster than $1/\sqrt{n}$ and hence is asymptotically negligible. Thus we get:

$$\mathbb{E}[f_2(x) - f_*(x)] = O\left(\kappa(\sqrt{V_2}, \mathcal{F}_*(\mu_n)) + \sqrt{\frac{V_2 \log(1/\delta)}{n}}\right)$$

446    The expected regret bound follows by standard arguments by simply integrating the above high
447    probability bound.        □

448    Going back to our policy learning problem, let $x = (y, a, z)$ and:
$$v_{DR}(x; \pi) = \langle \theta_{DR}(y, a, z), \phi(\pi(z), z) \rangle \qquad (32)$$
449    be the doubly robust proxy value at every sample $x$ and policy $\pi$. Then we can apply this general
450    theorem to the policy learning problem where, $x = (y, a, z)$ and function space:
$$\mathcal{F}_\Pi = \{-v_{DR}(\cdot; \pi) : \pi \in \Pi\} \qquad (33)$$
451    Then Theorem 6 yields the following corollary:

452    **Corollary 7** (Variance-Based Policy Regret). *Let* $\pi_* = \arg\max_{\pi \in \Pi} \mathbb{E}[v_{DR}(x; \pi)]$, $r =$
453    $\sup_{\pi \in \Pi} \sqrt{\mathbb{E}[v_{DR}(z; \pi)^2]}$, $\mu_n = \Theta\left(\kappa(r, \mathcal{F}_\Pi) + r\sqrt{\frac{\log(1/\delta)}{n}}\right)$ *and*

$$V_2 = \sup_{\pi \in \Pi: \mathbb{E}[v_{DR}(x; \pi_*) - v_{DR}(x; \pi)] \leq \mu_n} Var(v_{DR}(x; \pi) - v_{DR}(x; \pi_*)). \qquad (34)$$

454    *Then the policy $\pi_2$ returned by the out-of-sample regularized ERM, satisfies w.p.* $1 - \delta$ *over the*
455    *randomness of $S$:*

$$\mathbb{E}[v_{DR}(\pi_*) - v_{DR}(\pi_2)] = O\left(\kappa(\sqrt{V_2}, \mathcal{F}_\Pi) + \sqrt{\frac{V_2 \log(1/\delta)}{n}}\right) \qquad (35)$$

456    *and expected regret* $O\left(\kappa(\sqrt{V_2}, \mathcal{F}_\Pi) + \sqrt{\frac{V_2}{n}}\right)$.

457    To arrive at our final theorem, we also need to account for the difference between $\mathbb{E}[v_{DR}(x; \pi)]$ and
458    $V(\pi)$. This difference essentially stems from the error in the nuisance estimates, which introduce an
459    error in $\theta_{DR}(y, a, z)$, such that $\mathbb{E}[\theta_{DR}(y, a, z) \mid z] \neq \theta(z)$. However, we can invoke the orthogonality
460    of the doubly robust estimator and the general theorem of [9] on generalization bounds of orthogonal
461    losses to get:

462    **Lemma 8.** *For any policy $\pi_0 \in \Pi$, let $\hat{\pi}$ be the outcome of any possibly randomized algorithm that*
463    *satisfies w.p. $1 - \delta/2$ a regret bound on the doubly robust objective, i.e. $\mathbb{E}[v_{DR}(x; \pi_0) - v_{DR}(x; \hat{\pi})] \leq$*
464    $R_{n,\delta}$. *Moreover, suppose that the nuisance estimates satisfy a mean-squared error bound*

$$\max\left\{\mathbb{E}[(\hat{\theta}(z) - \theta_0(z))^2], \mathbb{E}[\|\hat{\Sigma}(z) - \Sigma_0(z)\|_{Fro}^2]\right\} := \chi_n^2 \qquad (36)$$

465    *Then w.p. $1 - \delta$ over the randomness of the policy sample:*
$$V(\pi_0) - V(\hat{\pi}) \leq O\left(R_{n,\delta} + \chi_n^2\right) \qquad (37)$$

*Proof.* By Lemma 3 we have that the loss function $-\mathbb{E}[v_{DR}(x;\pi)]$ is universally orthogonal as defined in [9]. Moreover, the loss is smooth with respect to the outputs of the nuisance functions and hence the second order derivatives of the loss with respect to the outputs of the nuisance functions are bounded. Thus the lemma follows by Theorem 2 of [9]. $\qquad\square$

If we assume that the nuisance estimation algorithm guarantees that w.p. $1-\delta$, $\chi_n^2 \leq h_{n,\delta}^2$ then observe that combining Corollary 7 and Lemma 8, we get that for any policy $\pi_0$, the policy $\pi_2$ of the out-of-sample regularized ERM satisfies, w.p. $1-\delta$:

$$V(\pi_0) - V(\pi_2) \leq O\left(\kappa(\sqrt{V_2}, \mathcal{F}_\Pi) + \sqrt{\frac{V_2 \log(1/\delta)}{n}} + h_{n,\delta}^2\right)$$

Similarly, if we assume that the nuisance esitmation algorithm satisfies $\mathbb{E}[\chi_n^2] \leq h_n^2$, then:

$$\mathbb{E}[V(\pi_0) - V(\pi_2)] \leq O\left(\kappa(\sqrt{V_2}, \mathcal{F}_\Pi) + \sqrt{\frac{V_2 \log(1/\delta)}{n}} + h_n^2\right)$$

We continue by proving the probabilistic regret bound of the theorem and the in-expectation bound follows analogously.

Finally, we need to account for the error introduced by the nuisance errors on the quantity $V_2$, so as to connect it with the semi-parametric efficiency variance of each policy, i.e.:

$$\mathrm{Var}(v_{DR}^0(x;\pi)) \tag{38}$$

where $v_{DR}^0(x;\pi) = \langle \theta_{DR}^0(y,a,z), \phi(\pi(z),z)\rangle$, and $\theta_{DR}^0(y,a,z)$ is the analogue of the doubly robust function, $\theta_{DR}(y,a,z)$, evaluated at the true nuisance functions. Moreover, we want our the "regret slice" to be with respect to the true regret, i.e. we want to depend on the variance of policies that satisfy:

$$V(\pi_*^0) - V(\pi) := \mathbb{E}[v_{DR}^0(x;\pi_*^0) - v_{DR}^0(x;\pi)] \leq \mu_n' \tag{39}$$

where $\pi_*^0 = \arg\max_{\pi\in\Pi} V(\pi)$. We prove such a property in the following lemma:

**Lemma 9.** *Consider the setting of Corollary 7. Suppose that the mean squared error of the nuisance estimates is upper bounded w.p. $1-\delta$ by $h_{n,\delta}^2$ and let $\epsilon_n = \mu_n + h_{n,\delta}^2$. Then:*

$$V_2^0 = \sup_{\pi,\pi'\in\Pi_*(\epsilon_n)} \mathrm{Var}(v_{DR}^0(x;\pi) - v_{DR}^0(x;\pi')) \tag{40}$$

*Then $V_2 \leq V_2^0 + O(h_{n,\delta})$.*

*Proof.* First observe that by Lemma 8 with $\pi_0 = \pi_*$ and $\hat{\pi} = \pi$ (for any $\pi \in \mathcal{F}_\Pi^2$), we have that:

$$\mathbb{E}[v_{DR}(\pi_*) - v_{DR}(\pi)] \leq \mu_n \implies V(\pi_*) - V(\pi) \leq \mu_n + O(h_{n,\delta}^2)$$

Similarly if we let $\pi_*^0 = \arg\max_{\pi\in\Pi} \mathbb{E}[v_{DR}^0(x;\pi)] := V(\pi)$, then observe that by definition of $\pi_*$: $\mathbb{E}[v_{DR}(x;\pi_*^0) - v_{DR}(x;\pi_*)] \leq 0$. Thus applying again Lemma 8 with $\pi_0 = \pi_*^0$ and $\hat{\pi} = \pi_*$:

$$\mathbb{E}[v_{DR}(\pi_*^0) - v_{DR}(\pi_*)] \leq 0 \implies V(\pi_*^0) - V(\pi_*) \leq O(h_{n,\delta}^2)$$

Let $\Pi_*^0(\epsilon) = \{\pi \in \Pi : V(\pi_*^0) - V(\pi) \leq \epsilon\}$ and let $\epsilon_n = O(\mu_n + h_{n,\delta}^2)$. Thus we have that:

$$V_2 \leq \sup_{\pi\in\Pi_*(\epsilon_n)} \mathrm{Var}(v_{DR}(x;\pi) - v_{DR}(x;\pi_*))$$

Moreover, observe that $\pi_* \in \Pi_*^0(\epsilon_n)$. Hence:

$$V_2 \leq \sup_{\pi,\pi'\in\Pi_*(\epsilon_n)} \mathrm{Var}(v_{DR}(x;\pi) - v_{DR}(x;\pi'))$$

Moreover, by Lipschitzness of $\theta_{DR}(y,a,z)$ on the output of the nuisance functions, we also have that for any $\pi,\pi' \in \Pi(\epsilon_n)$:

$$\mathrm{Var}(v_{DR}(x;\pi) - v_{DR}(x;\pi')) \leq \mathrm{Var}(v_{DR}^0(x;\pi) - v_{DR}^0(x;\pi')) + O(h_{n,\delta}) \tag{41}$$

493 Hence, if we denote with:

$$V_2^0 = \sup_{\pi, \pi' \in \Pi_*(\epsilon_n)} \mathrm{Var}(v_{DR}^0(x; \pi) - v_{DR}^0(x; \pi'))$$

494 Then we conclude that:

$$V_2 = V_2^0 + O(h_{n,\delta})$$

495 $\qquad\qquad\qquad\qquad\qquad\qquad\qquad\qquad\qquad\qquad\qquad\qquad\qquad\qquad\square$

496 Invoking Lemma 9 and the concavity of the entropy integral function we get:

$$V(\pi_*^0) - V(\hat{\pi}) \leq O\left(\kappa(\sqrt{V_2^0}, \mathcal{F}_\Pi) + \sqrt{\frac{V_2^0 \log(1/\delta)}{n}} + h_{n,\delta}^2 + h_{n,\delta}\frac{1}{\sqrt{n}}\right) \qquad (42)$$

497 Since $h_{n,\delta} = o(1)$, the last term is of lower order. This concludes the proof of the main regret
498 Theorem 1.

## C  Review of Semi-parametric Efficiency Bounds

500 In this section, we review the theory of semi-parametric efficiency bounds studied in [16] and [3].

### C.1  Definitions

502 **Definition 1** (Mean Square Differentiability). *Let $f(x; \eta)$ denote the probability density function*
503 *of a random variable $x$ where $\eta \in H$ is a finite dimensional parameter. $f(x; \eta)^{1/2}$is $\mu$-mean*
504 *square continuously differentiable with respect to $\eta$ on $H$ with derivative $f_\eta(x; \eta)$ if for each $\eta \in H$*
505 *$\int \|f_\eta(x; \eta)\|^2 d\mu$ is finite, and for every $\eta_i \to \eta$ with $\int \|f_\eta(x; \eta_i) - f_\eta(x; \eta)\|^2 d\mu \to 0$*

$$\int \left(f(x; \eta_i)^{1/2} - f(x; \eta)^{1/2} - f_\eta(x; \eta)'(\eta_i - \eta)\right)^2 d\mu / \|\eta_i - \eta\|^2 \to 0$$

506 **Definition 2** (Smoothness). *$f(x; \eta)$ is smooth if (i) $\eta \in H$, $H$ is open; (ii) there is a measure $\mu$*
507 *dominating $f(x; \eta)$ for $\eta \in H$ such that $f(x; \eta)$ is continuous on $H$ a.s. $\mu$ ; (iii) $f(x; \eta)^{1/2}$ is mean*
508 *square differentiable.*

**Definition 3** (Score and Information Matrix). *For smooth $f(x; \eta)$ the score for $\eta$ is defined as*

$$S_\eta(x; \eta) := 2\frac{f_\eta(x; \eta)}{f(x; \eta)}$$

*in the support of $x$ and the information matrix is*

$$\mathcal{I}(\eta) = \int S_\eta S_\eta' f(x; \eta) d\mu.$$

509 **Definition 4** (Regularity). *A likelihood function $f(x; \eta)$, $\eta \in H$, is regular if it is smooth and*
510 *information matrix is non-singular in $H$. The efficiency bound of a regular model is given by*
511 *Cramer-Rao bound and equals $\mathcal{I}(\eta)^{-1}$.*

512 **Definition 5** (Linearity). *Define a set $\mathcal{T}$ to be linear if $as_1 + bs_2 \in \mathcal{T}$ for all real scalars $a$ and $b$*
513 *and elements $s_1$ and $s_2$ of $\mathcal{T}$.*

### C.2  Derivation of the Efficiency Bound

515 Let data $(x_1, \ldots, x_n)$ consist of i.i.d copies of the random vector $(y, a, z)$. A semi-parametric model
516 consists of a parameter vector $\alpha$ and a set of restrictions on the joint behavior of observables. In our
517 model, the restrictions are given by the first order conditions of the linear projection

$$\mathbb{E}\left[(y - \langle \theta_0(z), \phi(a, z) \rangle)\phi(a, z) \mid z\right] = 0$$

518 and the parameter is

$$\alpha = \int \langle \theta(z), \phi(\pi(z), z) \rangle f(z) dz$$

519 where $f(z)$ denotes the probability distribution function of $z$. First, we provide the definition of a
520 parametric submodel.

521 **Definition 6** (Parametric Submodel). *For estimators with i.i.d data, a parametric submodel corre-*
522 *sponds to a parameter vector $\eta$ and a likelihood function $\ell(x|\eta)$ for a single observation that satisfies*
523 *the semi-parametric restrictions.*

524 A parametric submodel is a subset of the model distributions satisfying the semi-parametric assump-
525 tions. The reason parametric submodels are useful in analyzing semi-parametric efficiency is that
526 for parametric models, the Cramer-Rao bound gives the lower bound on the variance of estimators
527 of a parameter under some regulatory conditions. Since semi-parametric models impose weaker
528 restrictions than any parametric model, it is natural to expect that the asymptotic variance of a
529 semi-parametric model is no smaller than the bound for the parametric model.

530 In a parametric submodel, our parameter of interest can be written as

$$\alpha = \int \langle \theta(z;\eta), \phi(\pi(z), z) \rangle f(z;\eta) dz \tag{43}$$

531 Next, we define the semi-parametric efficient bounds.

532 **Definition 7** (Semi-parametric Efficiency Bound). *The semi-parametric efficiency bound of a semi-*
533 *parametric estimator is defined as the supremum of the Cramer-Rao bounds for all regular parametric*
534 *submodels.*

535 This definition is intuitive because any semi-parametric estimator that is consistent and asymptotically
536 normal cannot have a lower variance than the supremum of Cramer-Rao bounds. The regulatory
537 conditions defined in Section C.1 guarantee that the Cramer-Rao bound is well-defined and gives an
538 asymptotic efficiency bound.

539 To be able to obtain the Cramer-Rao bound for the parameter of interest under a parametric submodel,
540 the parameter must be pathwise differentiable.

541 **Definition 8** (Pathwise Differentiability). *A parameter $\alpha$ is pathwise-differentiable if $\alpha(\eta)$ is differ-*
542 *entiable for all smooth parametric submodels and there exists $q \times 1$ random vector $d$ such that $\mathbb{E}[d'd]$*
543 *is finite and for all regular parametric submodels*

$$\frac{\partial \alpha(\eta_0)}{\partial \eta} = \mathbb{E}[dS'_\eta]$$

544 *where $\eta_0$ denotes the true value of the parameter in the sense that $\ell(x|\eta_0)$ corresponds to the*
545 *likelihood function that generates the data.*

546 Pathwise differentiability of a parameter is a weak condition because, by Riesz representation
547 theorem, a parameter is pathwise-differentiable if it can be written as a functional that is mean-square
548 continuous. From the definition of $\alpha$ in Equation (43) it is easy to see that $\alpha$ is pathwise-differentiable
549 by Riesz representation theorem.

550 For a pathwise-differentiable parameter, the Cramer-Rao bound can be written as a function of the
551 pathwise derivative using the Delta method.

$$
\begin{aligned}
\mathrm{Var}(\alpha(\eta_0)) &= \frac{\partial \alpha(\eta_0)}{\partial \eta} (\mathbb{E}[S_\eta S'_\eta])^{-1} \frac{\partial \alpha(\eta_0)}{\partial \eta}' \\
&= \mathbb{E}[dS'_\eta](\mathbb{E}[S_\eta S'_\eta])^{-1}\mathbb{E}[S_\eta d']
\end{aligned}
$$

552 We can write $\mathrm{Var}(\alpha(\eta))$ as a second moment of a random variable as follows

$$
\begin{aligned}
\mathrm{Var}(\alpha(\eta_0)) &= \mathbb{E}[dS'_\eta](\mathbb{E}[S_\eta S'_\eta])^{-1}\mathbb{E}[S_\eta d'] \\
&= \mathbb{E}\big[\mathbb{E}[dS'_\eta](\mathbb{E}[S_\eta S'_\eta])^{-1}S_\eta S'_\eta(\mathbb{E}[S_\eta S'_\eta])^{-1}\mathbb{E}[S_\eta d']\big] \\
&= \mathbb{E}[d_\eta d'_\eta]
\end{aligned}
$$

553 Note that $d_\eta$ is mean-zero since

$$
\begin{aligned}
\mathbb{E}[d_\eta] &= \mathbb{E}\big[\mathbb{E}[dS'_\eta](\mathbb{E}[S_\eta S'_\eta])^{-1}S_\eta\big] \\
&= \mathbb{E}[dS'_\eta](\mathbb{E}[S_\eta S'_\eta])^{-1}\mathbb{E}[S_\eta] \\
&= 0
\end{aligned}
$$

This is useful because the Cramer-Rao bound of $\alpha$ under a parametric submodel equals the variance of $d_\eta$. Note further from the definition of $d_\eta$ that it is the linear projection of pathwise-derivative $d$ on score $S_\eta$. Therefore, the largest value of this projection can be obtained by considering the projection space as the scores corresponding to all parametric submodels. To formalize this, we next define the tangent set:

**Definition 9** (Tangent Set). *Define the tangent set $\mathcal{T}$ to be the mean square closure of all $q$-dimensional linear combinations of scores $S_\eta$ for smooth parametric submodels:*

$$T = \{s \in \mathbb{R} : \mathbb{E}[\|s\|^2] \leq \infty, \quad \exists A_j S_{\eta j} \quad with \quad \lim_{j \to \infty} \mathbb{E}[\|s - A_j S_{\eta j}\|^2] = 0\}$$

The projection of $d$ on the tangent set should have a larger variance than any particular submodel, suggesting that the projection should give the semi-parametric efficiency bound. The mathematical meaning of this projection on the tangent set is a least-squares projection in a Hilbert space of random vectors. This projection is defined as:

$$\delta \in \mathcal{T}, \quad \mathbb{E}[(d - \delta)s] = 0 \quad \text{for all} \quad s \in \mathcal{T}$$

If $\mathcal{T}$ is linear, then $\delta$ exists and unique. It is called the efficient score because it equals the efficient influence function in asymptotically linear estimators.

**Theorem 10** ([16], Theorem 3.1). *Suppose that the parameter is differentiable, $\mathcal{T}$ is linear, and $\mathbb{E}[\delta\delta']$ is nonsingular, for the projection $\delta$ of $d$ on $\mathcal{T}$. Then semi-parametric efficiency bound equals $\mathbb{E}[\delta\delta']$.*

# D  Proof of Theorem 2

*Proof.* We follow the steps outlined in Section (C.2) to calculate the semi-parametric efficiency bound of the parameter of interest:

$$\alpha := \mathbb{E}[\langle \theta_0(z), \phi(\pi(z), z) \rangle] \tag{44}$$

Let $f(y, a \mid z)$ and $f(z)$ denote the conditional distribution of $(y, a)$ given $z$ and the marginal distribution of $z$, respectively. The density of data $(y, a, z)$ is then equal to:

$$f(y, a, z) = f(y, a \mid z)f(z)$$

We consider a regular parametric submodel, parameterized by $\eta$, to calculate the pathwise derivative of $\alpha(\eta)$:

$$f(y, a, z; \eta) = f(y, a \mid z; \eta)f(z; \eta)$$

The corresponding scores for the parametric submodel is given by:

$$s_\eta(y, a, z; \eta) = s_\eta(y, a \mid z; \eta) + s_\eta(z; \eta)$$

where $s_\eta(y, a, z; \eta) = 2\dfrac{f_\eta(y, a, z; \eta)}{f(y, a, z; \eta)}$, and other scores are defined similarly.

Under the parametric submodel $\alpha$ can be written as:

$$\alpha(\eta) = \int \langle \theta(z; \eta), \phi(\pi(z), z) \rangle f(z; \eta) dz \tag{45}$$

The first step in semi-parametric efficiency bound derivation is to show that $\alpha(\eta)$ is pathwise differentiable, i.e. there exists $d(y, a, z; \eta_0)$ such that

$$\frac{\partial \alpha(\eta)}{\partial \eta} = \mathbb{E}[d(y, a, z; \eta)S_\eta(y, a, z; \eta)]$$

Let $\eta_0$ denote the true parameter value in the sense that $f(y, a, z; \eta_0)$ corresponds to the density of the data. To show pathwise differentiability, we differentiate Equation (45) under the integral sign and evaluate at $\eta = \eta_0$:

$$\frac{\partial \alpha(\eta_0)}{\partial \eta} = \int \left\langle \frac{\partial \theta(z; \eta_0)}{\partial \eta}, \phi(\pi(z), z) \right\rangle f(z; \eta_0) dz + \int \langle \theta_0(z; \eta_0), \phi(\pi(z), z) \rangle \frac{\partial f(z; \eta_0)}{\partial \eta} dz \tag{46}$$

$$= \mathbb{E}\left[ \left\langle \frac{\partial \theta(z; \eta_0)}{\partial \eta}, \phi(\pi(z), z) \right\rangle \right] + \mathbb{E}[\langle \theta(z; \eta_0), \phi(\pi(z), z) \rangle s_\eta(z; \eta_0)] \tag{47}$$

To calculate $\partial\theta(z; \eta_0)/\partial\eta$ inside the expectations we use the first order conditions of the linear projection:

$$\mathbb{E}\left[(y - \langle\theta_0(z), \phi(a, z)\rangle)\phi(a, z) \mid z\right] = 0$$

$$\int (y - \langle\theta(z; \eta_0), \phi(a, z)\rangle)\phi_i(a, z)f(y, a \mid z; \eta_0)dyda = 0$$

Taking the derivative under the integral sign and evaluating at $\eta_0$ for all $i$:

$$\mathbb{E}\left[\left\langle\frac{\partial\theta(z; \eta_0)}{\partial\eta}, \phi(a, z)\phi(a, z)^T\right\rangle \mid z\right] + \mathbb{E}[(y - \langle\theta(z; \eta_0), \phi(a, z)\rangle)\phi(a, z)s_\eta(y, a \mid z, \eta_0) \mid z] = 0$$

Solving for $\partial\theta(z; \eta_0)/\partial\eta$

$$\partial\theta(z; \eta_0)/\partial\eta = \mathbb{E}\left[\Sigma(z)^{-1}\phi(a, z)(y - \langle\theta(z; \eta_0), \phi(a, z)\rangle)s_\eta(y, a \mid z; \eta_0) \mid z\right]$$

Substituting this into Equation (47):

$$\frac{\partial\alpha(\eta_0)}{\partial\eta} = \mathbb{E}\left[\left\langle\Sigma_0(z)^{-1}\phi(a, z)(y - \langle\theta_0(z), \phi(a, z)\rangle), \phi(\pi(z), z)\right\rangle s_\eta(y, a \mid z; \eta_0)\right] + \qquad (48)$$

$$\mathbb{E}[\langle\theta_0(z), \phi(\pi(z), z)\rangle s_\eta(z; \eta_0)]$$

$$= \mathbb{E}\left[\left(\langle\theta_0(z) + \Sigma_0(z)^{-1}\phi(a, z)(y - \langle\theta_0(z), \phi(a, z)\rangle), \phi(\pi(z), z)\rangle - \alpha(\eta_0)\right)(s_\eta(y, a \mid z, \eta_0) + s_\eta(z; \eta_0))\right]$$

$$= \mathbb{E}\left[d(y, a, z; \eta_0)(s_\eta(y, a \mid z; \eta_0) + s_\eta(z; \eta_0))\right]$$

$$= \mathbb{E}\left[d(y, a, z; \eta_0)(s_\eta(y, a \mid z; \eta_0))\right] \qquad (49)$$

The second line follows because:

$$\mathbb{E}[\langle\theta_0(z), \phi(\pi(z), z)\rangle s_\eta(y, a \mid z, \eta_0)] = \mathbb{E}[\langle\theta_0(z), \phi(\pi(z), z)\rangle\mathbb{E}[s_\eta(y, a \mid z, \eta_0) \mid z]] = 0$$

$$\mathbb{E}[\alpha(\eta_0)s_\eta(z; \eta_0)] = \alpha(\eta_0)\mathbb{E}[s_\eta(z; \eta_0)] = 0$$

and

$$\mathbb{E}[\langle\Sigma_0(z)^{-1}\phi(a, z)(y - \langle\theta_0(z), \phi(a, z)\rangle), \phi(\pi(z), z)\rangle s_\eta(z; \eta_0)] = 0$$

Subtracting $\alpha(\eta_0)$ in the second line makes the pathwise derivative mean zero, which will prove useful later when projecting $d(y, a, z; \eta_0)$ on the tangent set.

Since Equation (48) satisfies the condition given in the defition of pathwise differentiability, the pathwise derivative of $\alpha(\eta)$ is:

$$d(y, a, z; \eta_0) = \left(\langle\theta_0(z) + \Sigma_0(z)^{-1}\phi(a, z)(y - \langle\theta_0(z), \phi(a, z)\rangle), \phi(\pi(z), z)\rangle - \alpha\right)$$

The semi-parametric efficiency bound for $\alpha$ is the variance of the projection of $d(y, a, z; \eta_0)$ onto the tangent space defined as the closed linear span of the scores:

$$\mathcal{T} = \{s(y, a \mid z) + s(z)\}$$

Note that the joint distribution is unrestricted so the only restrictions on the score functions are $E[s(y, x \mid z) \mid z] = 0$ and $E[s(z)] = 0$ and they are smooth.

Next, we show that the pathwise derivative is already in the tangent set $d(y, a, z; \eta_0) \in \mathcal{T}$. To see this we can write $d(y, a, z; \eta_0)$ as the sum of two functions:

$$d(y, a, z; \eta_0) = \left(\Sigma_0(z)^{-1}\phi(a, z)(y - \langle\theta_0(z), \phi(a, z)\rangle), \phi(\pi(z), z)\rangle\right) + \left(\langle\theta_0(z), \phi(\pi(z), z)\rangle - \alpha\right)$$

The first component is mean independent of $z$:

$$\mathbb{E}[\langle\Sigma_0(z)^{-1}\phi(a, z)(y - \langle\theta_0(z), \phi(a, z)\rangle), \phi(\pi(z), z)\rangle \mid z] = 0$$

The second component is function of only $z$ and has zero mean:

$$\mathbb{E}[\langle\theta_0(z), \phi(\pi(z), z)\rangle - \alpha] = 0$$

Therefore, the pathwise derivative equals the sum of two functions that satisfy the restrictions on score functions in the tangent set, namely, $E[s(y, x \mid z) \mid z] = 0$ and $E[s(z)] = 0$. From this, we

conclude that $d(y, a, z; \eta_0)$ is in the tangent set; so the projection of $d(y, a, z; \eta_0)$ onto $\mathcal{T}$ is equal to itself.

Therefore, the efficiency bound for $\alpha$ is:

$$
\begin{aligned}
V_{eff}(\alpha) &= Var(d(y, a, z; \eta_0)) \\
&= Var(v_{DR}(y, a, z; \pi))
\end{aligned}
$$

Therefore, the doubly robust estimator, $v_{DR}(y, a, z; \pi)$, achieves the semi-parametric efficiency bound. This result extends to the difference of value functions by linearity of pathwise derivative.

To investigate the semi-parametric efficiency bound under the correct specification we use a result from [4] who shows that under the correct specification the efficiency bound is:

$$
\begin{aligned}
V_{eff}^c(\alpha) &= Var\big(\langle \theta_0(z), \phi(\pi(z), z) \rangle\big) \\
&\quad + \mathbb{E}\big[\phi(\pi(z), z)\mathbb{E}[\phi(a, z)\mathbb{E}[\epsilon^2 \mid a, z]^{-1}\phi(a, z)' \mid z]^{-1}\phi(\pi(z), z)^T\big]
\end{aligned}
$$

where $\epsilon = (y - \langle \theta_0(z), \phi(a, z) \rangle)$ is defined as residuals.

Under the homoskedastivity assumption, $\mathbb{E}[\epsilon^2 \mid a, z] = \sigma^2$, this efficiency bound becomes:

$$
\begin{aligned}
V_{eff}^c(\alpha) &= Var\big(\langle \theta_0(z), \phi(\pi(z), z) \rangle\big) + \\
&\quad \sigma^2 \mathbb{E}\big[\phi(\pi(z), z)\mathbb{E}[\phi(a, z)\phi(a, z)' \mid z]^{-1}\phi(\pi(z), z)^T\big] \\
&= Var\big(\langle \theta_0(z), \phi(\pi(z), z) \rangle\big) + \sigma^2 \mathbb{E}\big[\phi(\pi(z), z)\Sigma_0(z)^{-1}\phi(\pi(z), z)^T\big]
\end{aligned}
$$

which is equal to the variance of the doubly robust estimator:

$$
\begin{aligned}
V_{eff}(\alpha) &= Var\big(\langle \theta_0(z), \phi(\pi(z), z) \rangle\big) + \\
&\quad \mathbb{E}\big[\phi(\pi(z), z)\mathbb{E}[\Sigma_0(z)^{-1}\phi(a, z)\epsilon^2\phi(a, z)'\Sigma(z)^{-1} \mid z]\phi(\pi(z), z)^T\big] \\
&= Var\big(\langle \theta_0(z), \phi(\pi(z), z) \rangle\big) + \\
&\quad \sigma^2 \mathbb{E}\big[\phi(\pi(z), z)\Sigma_0(z)^{-1}\mathbb{E}[\phi(a, z)\phi(a, z)' \mid z]\Sigma(z)^{-1}\phi(\pi(z), z)^T\big] \\
&= Var\big(\langle \theta_0(z), \phi(\pi(z), z) \rangle\big) + \\
&\quad \sigma^2 \mathbb{E}\big[\phi(\pi(z), z)\Sigma_0(z)^{-1}\Sigma_0(z)^{-1}\Sigma_0(z)^{-1}\phi(\pi(z), z)^T\big] \\
&= Var\big(\langle \theta_0(z), \phi(\pi(z), z) \rangle\big) + \sigma^2 \mathbb{E}\big[\phi(\pi(z), z)\Sigma_0(z)^{-1}\phi(\pi(z), z)^T\big] \\
&= Var(v_{DR}(y, a, z; \pi))
\end{aligned}
$$

$\square$

# E Double Robustness Property of Policy Estimator

**Theorem 11** (Double Robustness). $V_{DR}(\pi)$ *is an unbiased estimate of* $V_0(\pi(z), z)$ *if for all $z$, either* $\mathbb{E}_{S_1 \sim D^{n/2}}[\hat{\theta}(z)] = \theta_0(z)$ *or* $\mathbb{E}_{S_1 \sim D^{n/2}}[\hat{\Sigma}(z)^{-1}] = \Sigma_0(z)^{-1}$, *where expectation is taken over the randomness of the nuisance estimation sample $S_1$.*

*Proof.* Let $\bar{\theta}(z) = \mathbb{E}_{S_1 \sim D^{n/2}}[\hat{\theta}(z)]$ and $\bar{\Sigma}^{-1}(z) = \mathbb{E}_{S_1 \sim D^{n/2}}[\hat{\Sigma}(z)^{-1}]$, be the expected value of the estimates at any input $z$, where the expectation is with respect to the randomness on the half-split of $n/2$ samples that were used for training the estimates. Due to sample-splitting and cross-fitting, the expected value of the doubly robust policy estimate can be written as:

$$
\mathbb{E}[V_{DR}(\pi)] = \mathbb{E}\left[\langle \bar{\theta}(z) + \bar{\Sigma}(z)^{-1}\phi(a, z)(y - \langle \bar{\theta}(z), \phi(a, z) \rangle), \phi(\pi(z), z) \rangle\right] \tag{50}
$$

where the random variables $(y, a, z)$ are a fresh independent draw of the data generating process that generated the observational data.

Observe that $y$ is an unbiased estimate of $V(a, z)$ conditional on $z$. Moreover, since $\theta_0(z)$ is the minimizer of the conditional squared loss, taking the first order condition implies:

$$
\begin{aligned}
\mathbb{E}[(V_0(a, z) - \langle \theta_0(z), \phi(a, z) \rangle)\phi(a, z) \mid z] = 0 \iff \\
\mathbb{E}[y\,\phi(a, z) \mid z] = \mathbb{E}[\langle \theta_0(z), \phi(a, z) \rangle)\,\phi(a, z) \mid z]
\end{aligned}
$$

Thus we can re-write the expected value of the doubly robust policy estimate as:

$$
\begin{aligned}
\mathbb{E}[V_{DR}(\pi)] &= \ \mathbb{E}\left[\langle \bar{\theta}(z) + \bar{\Sigma}(z)^{-1}\,\phi(a,z)\,(Y - \langle\bar{\theta}(z), \phi(a,z)\rangle)\,, \phi(\pi(z),z)\rangle\right] \\
&= \ \mathbb{E}\left[\langle \bar{\theta}(z) + \bar{\Sigma}(z)^{-1}\,\phi(a,z)\,\langle\theta_0(z) - \bar{\theta}(z), \phi(a,z)\rangle, \phi(\pi(z),z)\rangle\right] \\
&= \ \mathbb{E}\left[\langle \bar{\theta}(z) + \bar{\Sigma}(z)^{-1}\,\phi(a,z)\phi(a,z)^T(\theta_0(z) - \bar{\theta}(z)), \phi(\pi(z),z)\rangle\right] \\
&= \ \mathbb{E}\left[\langle \bar{\theta}(z) + \bar{\Sigma}(z)^{-1}\,\mathbb{E}[\phi(a,z)\phi(a,z)^T \mid z](\theta_0(z) - \bar{\theta}(z)), \phi(\pi(z),z)\rangle\right] \\
&= \ \mathbb{E}\left[\langle \bar{\theta}(z) + \bar{\Sigma}(z)^{-1}\,\Sigma_0(z)(\theta_0(z) - \bar{\theta}(z)), \phi(\pi(z),z)\rangle\right] \\
&= \ \mathbb{E}\left[\langle \bar{\theta}(z) + \bar{\Sigma}(z)^{-1}\,\Sigma_0(z)(\theta_0(z) - \bar{\theta}(z)), \phi(\pi(z),z)\rangle\right]
\end{aligned}
$$

Hence we have:

$$
\mathbb{E}[V_{DR}(\pi)] - V_0(\pi) = \ \mathbb{E}\left[\langle \left(\bar{\Sigma}(z)^{-1}\Sigma_0(z) - \mathbb{I}\right)\left(\theta_0(z) - \bar{\theta}(z)\right), \phi(\pi(z),z)\rangle\right]
$$

The right hand side is zero if either $\bar{\theta}(z) = \theta_0(z)$ or if $\bar{\Sigma}(z)^{-1} = \Sigma_0(z)$. $\qquad\square$

# F  Lipschitz Variogram Settings and Binary Treatment

For simplicity of notation, we let $v(x;\pi) = v_{DR}^0(x;\pi)$ and $\pi_* = \pi_*^0$ throughout this section, as the results are not specific to the doubly robust value function. Suppose that the value function of the policy learning problem has the following self-bounded Lipschitz property:

$$
\mathrm{Var}(v(x;\pi)) - C\,\mathrm{Var}(v(x;\pi_*)) \leq L\,|\mathbb{E}[v(x;\pi)] - \mathbb{E}[v(x;\pi_*)]| = L(V(\pi_*) - V(\pi))
$$

for some constants $C, L$, i.e. if a policy has value close to the optimal policy, the it does not have much larger variance. Then we have that:

$$
\begin{aligned}
\sup_{\pi,\pi'\in\Pi_*(\epsilon_n)} \mathrm{Var}(v(x;\pi) - v(x;\pi)) &\leq \sup_{\pi\in\Pi_*(\epsilon_n)} 4\,\mathrm{Var}(v(x;\pi)) \\
&\leq 4C\,\mathrm{Var}(v(x;\pi_*)) + 4\,L \sup_{\pi\in\Pi_*(\epsilon_n)} (V(\pi) - V(\pi_*)) \\
&\leq 4C\,\underbrace{\mathrm{Var}(v(x;\pi_*))}_{V_*} + 4\,L\,\epsilon_n
\end{aligned}
$$

Thus we get regret rates of the form:

$$
\begin{aligned}
V(\pi_*) - V(\pi_2) &= \ O\left(\kappa(2\sqrt{C\,V_*}, \mathcal{F}_\Pi) + \sqrt{\frac{V_*\,\log(1/\delta)}{n}} + \epsilon_n\frac{1}{\sqrt{n}}\right) \\
&= \ O\left(\kappa(2\sqrt{C\,V_*}, \mathcal{F}_\Pi) + \sqrt{\frac{V_*\,\log(1/\delta)}{n}}\right)
\end{aligned}
$$

since $\epsilon_n = o(1)$.

**Example 3** (Binary Treatment). *In the case of binary treatment considered in [1], the loss took the form:*

$$
v(x;\pi) = \Gamma(z)\cdot(2\pi(z) - 1) \tag{51}
$$

*with $\pi : Z \to \{0,1\}$. In this case observe that the self-bounded property is satisfied since:*

$$
\begin{aligned}
\mathit{Var}(v(x;\pi)) &= \ \mathbb{E}[v(x;\pi)^2] - \mathbb{E}[v(x;\pi)]^2 \\
&= \ \mathbb{E}[\Gamma(z)^2(2\pi(z) - 1)^2] - V(\pi)^2 \\
&= \ \mathbb{E}[\Gamma(z)^2] - V(\pi)^2
\end{aligned}
$$

*Where the latter property holds since $(2\pi(z) - 1)^2 = 1$ irrespective of $\pi(z)$. Thus the first part in the variance is independent of the policy, which is the crucial special property of the binary treatment case. This leads to the fact that:*

$$
\mathit{Var}(v(x;\pi)) - \mathit{Var}(v(x;\pi_*)) = V(\pi_*)^2 - V(\pi)^2 \leq 2\,|V(\pi) - V(\pi_*)| \tag{52}
$$

*Hence, the self-boundedness property holds with $C = 1$ and $L = 2$. Thus for the binary treatment setting we can achieve a regret rate whose leading term only depends on the semi-parametric efficient variance of the optimal policy.*

As a concrete example, consider the case when the class $\mathcal{F}_\Pi$ is a VC-subgraph class of VC dimension $d$, and let $S_n = \mathbb{E}_n[\sup_\pi v(x; \pi)^2] = \mathbb{E}_n[\Gamma(z)^2]$. Then Theorem 2.6.7 of [22] shows that: $\mathcal{H}_2(\epsilon, \mathcal{F}_\Pi, n) = O(d(1 + \log(S_n/\epsilon)))$. This implies that

$$\kappa(r, \mathcal{F}_\Pi) = O\left(\int_0^r \sqrt{d(1 + \log(S_n/\epsilon))}d\epsilon\right) = O\left(r\sqrt{d}\sqrt{1 + \log(S/r)}\right).$$

Moreover, by Markov's inequality w.p. $1 - \delta$, $S_n \leq \mathbb{E}[S_n]/\delta = \mathbb{E}[\sup_\pi v(x; \pi)^2]/\delta = \mathbb{E}[\Gamma(z)^2]/\delta := S/\delta$. Hence, we can conclude that w.p. $1 - \delta$:

$$V(\pi_*) - V(\pi_2) = O\left(r\sqrt{1 + \log(S/r)}\sqrt{\frac{d}{n}} + r\sqrt{\frac{\log(1/\delta)}{n}} + \frac{d(1 + \log(S/r))}{n}\frac{\log(1/\delta)}{n}\right).$$

Combining all the above we get a bound of the form (excluding lower order terms):

$$V(\pi_*) - V(\pi_2) = O\left(\sqrt{V_*(1 + \log(S/V_*))}\sqrt{\frac{d}{n}} + \sqrt{\frac{V_* \log(1/\delta)}{n}}\right).$$

which recovers the result of [1] up to constants.

# G   Application: Costly Resource Allocation

Motivated by a resource allocation scenario, we also analyze experimentally the special case where $\phi(a, z) = a$. Consider the case where we have $p$ possible tasks to invest in, and we have investment costs. Each task yields a return on investment that is a linear function of the investment, but an unknown function $\theta(z)$ of the context $z$. Moreover, to maintain an investment portfolio of $\pi(z)$ we need to pay a known cost $C(\pi(z))$. Given a policy space $\Pi : \mathcal{Z} \rightarrow \mathbb{R}^p$, our goal is to optimize:

$$\sup_{\pi \in \Pi} \mathbb{E}\left[\langle\theta(z), \pi(z)\rangle - C(\pi(z))\right] \tag{53}$$

This falls into our framework, if we treat the offset part as of the form $\langle\theta_0(z), C(\pi(z))\rangle$ but with a known $\theta_0(z) = 1$. So in that case we simply consider $\theta_{DR,0}(z) = \theta_0(z) = 1$. Then applying our framework we optimize:

$$\sup_{\pi \in \Pi} \mathbb{E}_n\left[\langle\theta_{DR}(z), \pi(z)\rangle - C(\pi(z))\right] \tag{54}$$

In the case of quadratic costs $C(\pi(z)) = \frac{\lambda}{2}\|\pi(z)\|_2^2$, then this boils down to exactly optimizing a square loss objective, since:

$$\inf_A \mathbb{E}_n\left[\|\theta_{DR}(z)/\lambda - \pi(z)\|^2\right] \Leftrightarrow \sup_A \mathbb{E}\left[\langle\theta_{DR}(z), \pi(z)\rangle\right] - \frac{\lambda}{2}\mathbb{E}_n\left[\|\pi(z)\|_2^2\right] \tag{55}$$

Thus policy optimization reduces to a multi-task regression problem where we are trying to predict $\theta_{DR}(z)/\lambda$ from $z$.[6]

We can consider sparse linear policies:

$$\Pi = \{z \rightarrow Az : \|A\|_{11} := \sum_i \|\alpha_i\|_1 \leq s\} \tag{56}$$

where $\alpha_i$ corresponds to the $i$-th row of matrix $A$. In this case our problem reduces to the MultiTask Lasso problem where the label is $\theta(z)/\lambda$.

**Experimental Evaluation.**   For experimental evaluation we consider a model with two tasks, $a_1$ and $a_2$:

$$y = a(z)a_1 + b(z)a_2 + \epsilon$$

(a) Low Dimensional Regime

(b) High Dimensional Regime

Figure 2: Costly Resource Allocation: Each line shows the mean and standard deviation of regret over 100 simulations.

We use the same distributions and functions, $a(z)$ and $b(z)$, given above for the pricing application. To estimate the optimal allocation and its regret, we run a 5-fold cross validated MultiTask Lasso algorithm and set $\lambda = 1$. We report the distribution of return on investment obtained from different models in Figure (2). The results suggest that doubly robust method achieves a significantly lower regret than the direct method in both regimes and its performance is similar to the oracle method [7].

# H  Doubly Robust Estimators in Pricing Experiment

## H.1  Linear Model

We want to estimate some regression models of $a(z)$ and $b(z)$ in the demand model. For instance, if these fall in some high-dimensional linear function class, we can estimate a regression between demand and the linear function class. Moreover, we need to estimate the covariance matrix, which in this case takes the simple form:

$$\Sigma_0(z) = \begin{bmatrix} 1 & \mathbb{E}[p \mid z] \\ \mathbb{E}[p \mid z] & \mathbb{E}[p^2 \mid z] \end{bmatrix} \tag{57}$$

whose inverse takes the form:

$$\Sigma_0(z)^{-1} = \frac{1}{\mathrm{Var}(p \mid z)} \begin{bmatrix} \mathbb{E}[p^2 \mid z] & -\mathbb{E}[p \mid z] \\ -\mathbb{E}[p \mid z] & 1 \end{bmatrix} \tag{58}$$

If for instance the observational policy was homoskedastic (i.e. the exploration component was independent of the context $z$), then $\mathrm{Var}(p \mid z)$ is a constant $\sigma^2$ independent of $z$. Moreover, we can write:

$$\mathbb{E}[p^2 \mid z] = \sigma^2 + \mathbb{E}[p \mid z]^2 \tag{59}$$

Thus we only need to estimate the mean treatment policy $g(z) = \mathbb{E}[p \mid z]$ and the variance $\sigma^2$. Then the doubly robust estimate of $a(z)$ takes the form:

$$a_{DR}(z) = \hat{a}(z) + \left( 1 + \hat{g}(z)\frac{\hat{g}(z) - p}{\hat{\sigma}^2} \right)(d - \hat{a}(z) - \hat{b}(z)\,p)$$

$$b_{DR}(z) = \hat{b}(z) + \frac{p - \hat{g}(z)}{\hat{\sigma}^2}(d - \hat{a}(z) - \hat{b}(z)\,p)$$

## H.2 Quadratic Model

In the case where we observe the revenue our model becomes quadratic in prices
$$r = a(z)x - b(z)x^2 + \epsilon$$

The covariance matrix takes the form:
$$\Sigma_0(z) = \begin{bmatrix} \mathbb{E}[p^2 \mid z] & \mathbb{E}[p^3 \mid z] \\ \mathbb{E}[p^3 \mid z] & \mathbb{E}[p^4 \mid z] \end{bmatrix}$$

whose inverse is:
$$\Sigma_0(z)^{-1} = \frac{1}{\mathbb{E}[p^4 \mid z]\mathbb{E}[p^2 \mid z] - \mathbb{E}[p^3 \mid z]^3} \begin{bmatrix} \mathbb{E}[p^4 \mid z] & -\mathbb{E}[p^3 \mid z] \\ -\mathbb{E}[p^3 \mid z] & \mathbb{E}[p^2 \mid z] \end{bmatrix}$$

Let $\mu_k(z)$ denote $E[p^k \mid z]$. If the observational policy was homoskedastic and none of the central moments of price depends on $z$, using the recursive structure, the nuisance functions in the covariance matrix can be written as

$$\mu_2(z) = \mu_2^c + \mu_1(z)^2$$
$$\mu_3(z) = \mu_3^c + 3\mu_2(z)\mu_1(z) - 2\mu_1(z)^3$$
$$\mu_4(z) = \mu_4^c + 4\mu_3(z)\mu_1(z) - 6\mu_1(z)\mu_2(z) + 3\mu_1(z)^4$$

where $\mu_k^c$ denotes the k-th central moment of $p$. Therefore, we only need to estimate the mean treatment policy $\mu_1(z)$ and the central moments $\mu_2^c$, $\mu_3^c$ and $\mu_4^c$. Then, the doubly robust estimate of $a(z)$ and $b(z)$ take the form:

$$a_{DR}(z) = \hat{a}(z) + \left( \frac{\mu_4(z)p - \mu_3(z)p^2}{\mu_4(z)\mu_2(z) - \mu_3(z)^2} \right) \left( d - \hat{a}(z)p - \hat{b}(z)\,p^2 \right)$$

$$b_{DR}(z) = \hat{b}(z) + \left( \frac{\mu_2(z)p^2 - \mu_3(z)p}{\mu_4(z)\mu_2(z) - \mu_3(z)^2} \right) \left( d - \hat{a}(z)p - \hat{b}(z)\,p^2 \right)$$

# I  Additional Experiment Results

(a) Policy Evaluation

(b) Regret

Figure 3: Linear, High Dimensional Regime: (a) Black line shows the true value of the policy, and each line shows the mean and standard deviation of the policy over 100 simulations. (b) each line shows the mean and standard deviation of the value of the corresponding policy over 100 simulations. We omit the results for the inverse propensity score method since they are too large to report together with the other estimates in the high dimensional regime.

(a) Policy Evaluation

(b) Regret

Figure 4: Quadratic, Low Dimensional Regime: (a) Black line shows the true value of the policy, each line shows the mean and standard deviation of the value of the corresponding policy over 100 simulations. (b) Each line shows the mean and standard deviation of regret over 100 simulations. We omit the results for the inverse propensity score method since they are too large to report together with the other estimates in the high dimensional regime.

(a) Policy Evaluation

Figure 5: Quadratic, High Dimensional Regime: (a) Black line shows the true value of the policy, and each line shows the mean and standard deviation of the policy over 100 simulations. (b) each line shows the mean and standard deviation of the value of the corresponding policy over 100 simulations. We omit the results for the inverse propensity score method since they are too large to report together with the other estimates in the high dimensional regime.

## Footnotes

[6]The above reasoning extends to heterogeneous costs across tasks e.g. $C(\pi(z)) = \sum_i c_i\pi_i(z)^2$. In this case the label of the $i$-th task of the multi-task regression problem is $\theta_{DR,i}(z)/c_i$ and we need to perform a weighted multi-task regression where the weight on the square loss for task $i$ is equal $c_i$.

[7]For comparison, the value achieved by best-in-class policy is 22.2 in low dimensional regime and ? in high dimensional regime. We omit the inverse propensity score regrets since they are too large to report together with other estimates