[Reviews · NeurIPS 2019]

Reviewer 1



This paper considers the off-policy learning problem for the case of continuous treatments, and provides regret bounds for the doubly-robust estimator, as well as study of semiparametric efficiency. The primary assumptions are that the “value function” is of known parametric form in the treatment, but with arbitrary dependence on covariates. In this sense, it generalizes the classical partially linear regression model. Originality: The paper generalizes recent semiparametric efficient analysis of doubly robust estimators, particularly the "slicing" analysis innovation for policy learning as in Athey and Wager, Efficient policy learning, as well as analysis of Foster and Syrgkanis 2019. The proposed approach for continuous treatments avoids the unfavorable dimension dependence of previous approaches for continuous treatments, instead the difficulty is in the matrix regression problem of the covariance-based generalization of the propensity score for the continuous case. Quality: The paper is technically sound with claims well supported by theoretical analysis. Clarity: The paper is overall clear but sometimes vague in descriptions. I found the remarks helpful in instantiating the results. Fig. 1a: the axes are unreadable and the results could be summarized differently or moved to the appendix. Significance: The paper studies continuous treatments for policy learning and applies recent advances in analysis for policy learning and doubly-robust estimators to this setting, in order to study the doubly-robust estimator. Minor comments/clarification questions: - Lines 93-98: could you clarify the relationship to the “slates” estimator of Swaminathan et al. 2017, as well as the dependence of the rate on the estimate of \hat\Sigma(z), or include a reference on a “typical” nuisance rate in that setting? - The efficiency bound is achieved if the model is misspecified, while requiring an additional homoskedasticity assumption otherwise if it is achieved: this seems nonstandard, perhaps commenting on why this arises inline could be useful. - Regarding remark 4 abstracting away technical details regarding first-stage rates: while this may hold for the outcome regression, for the matrix-valued regression of \hat\Sigma, specific details regarding typical rates and their dependence on dimension or other quantities may be helpful to the reader. For example, the special assumption of homoskedasticity for the pricing example simplifies the problem to simply that of density estimation; while convenient, this abstracts away some potential issues in implementing this approach in practice.

Reviewer 2



This is a very interesting work dealing with continuous action spaces, and develops associated semiparametric theory for this framework. In Theorem 2, why does V_{DR}(\pi) achieve the efficiency bound if the model is misspecified (contrary to the usual DR result so that we only get consistency but not efficiency)? Also, why do we need the additional assumption of homoskedasticity when the model is correct?

Reviewer 3



The paper builds on recent work on orthogonal machine learning, efficient policy learning, and continuous-action policy learning. The theory is solid and it is a reasonable advance with good empirical results. The results are most similar to Foster and Syrgkanis [9], which makes the present paper less novel and interesting. The big improvement over [9], per the authors, is that they provide an analysis giving a regret guarantee for the unpenalized policy learning approach that depends on the efficient variance of estimating the optimal policy. [9], in contrast, provided a bound that depended on the variance of estimating any policy in the class, and only show a regret bound depending on the variance of estimating the optimal policy for a modified learning procedure that penalizes the second moment. First, I'm not sure how groundbreaking improving the bound to depend on the efficient variance of the optimal policy instead of an arbitrary policy is. Sure, it's better, but is it of sufficient general interest to merit acceptance? Second, I'm not sure I buy the authors' claim about the computational issues of variance regularization. Optimizing the the orthogonal machine learning estimate over policies is already not convex and rather intractable (and the authors don't explain how they actually optimize it in section 4!). So adding a variance regularizer does not destroy any convexity. Also, unlike the policy optimization objective, which is difficult to convexify for continuous actions (where for binary actions one usually uses classification surrogate losses), the variance regularization has very nice convexifications via phi-divergences / distributionally robust optimization. Post-response: I have read the authors' response and I find that it appropriately and sufficiently addresses my questions about comparison to [9] and the tractability of the different optimization problems. But I would suggest the authors to add discussion reflecting the explanations they offered in the response into the text at camera ready.

[Author Response · NeurIPS 2019]

We thank the reviewers for their thoughtful comments and suggestions and we respond below to some concrete questions/comments that were raised.

**Response to Reviewer #4.** *Slate estimator.* We agree that we should have added a reference to the Swaminathan et al paper. The setup there is also a special case of our setup, where the reward is linear in the treatment vector, i.e. $\langle\theta(z), T\rangle$, but where $T \in R^{\ell \cdot m}$ ($\ell$ is number of items and $m$ number of slots) and where $T$ takes values in a subset of the hypercube. The discreteness of the action space allows Swaminathan et al to apply a more direct propensity approach (see e.g. Remark 2 for a similar example). Moreover, the slate estimator uses solely the IPS part and is not doubly robust, as the paper works in the setting with a known propensity function.

*Efficiency.* This phenomenon is rather standard in the econometrics literature: if one assumes both that the model is well-specified and that there is heteroskedastic noise, then one could typically construct more efficient estimators by optimally re-weighting the samples inversely proportionate to the variance of the error for the corresponding $z_i$. Such optimally re-weighted estimators are typically avoided in practice as they heavily rely on the well-specification of the model. Hence, we omitted such an analysis. We will add a relevant discussion in the revision with reference of results similar in flavor and an appendix section of how an optimally re-weighted estimator would look like.

*Estimating co-variance.* In the worst-case one can view the estimation of the co-variance as a set of separate regressions, one for each entry of the matrix (see e.g. Equation 6 and the sentence above). Assuming the matrix has small dimension and assuming some high-dimensional model space for each regression, then typical regression rates would apply. The example of pricing however shows that in natural problems one might be able to get away with even simpler estimators for this co-variance matrix. We will add some more elaborate discussion on these rates expanding on Remark 4.

**Response to Reviewer #5.** See response also to Reviewer #4 regarding efficiency. Roughly: if one assumes the model is well-specified then this implies many more moment conditions than just the unconditional moment implied by a square loss projection. These extra moments can be used to construct more efficient estimators. This can indeed lead to a benefit if the errors are heteroskedastic as then one should do an optimally re-weighted square loss projection. However, these extra moment conditions have no bite in the case of homoskedastic noise. The technical proof in appendix D goes through such an argument. We will add a sketch that highlights these main points.

**Response to Reviewer #6.** *Relation to Foster and Syrgkanis (FS).* Our paper does use the framework of (FS) and the main theorems in that work as a stepping stone in our regret results. However, there are two substantial contributions: 1) the framework of (FS) starts from the assumption that one has formulated an orthogonal loss, but does not provide any way of acquiring such an orthogonal loss. Hence, our first contribution is constructing an orthogonal loss in the case of policy learning with continuous actions. Existence of such orthogonal losses were left as an open question in prior work (e.g. Athey and Wager). 2) The out-of-sample regularized ERM provides both a computationally efficient alternative to the variance penalization whenever the original ERM problem is convex and also attains a regret bound whose leading term is much better than that achieved by variance or moment penalization: i) we get a bound that depends on the entropy integral of the policy space as opposed to the critical radius that was achieved by (FS), or the even worse metric entropy at $O(1/n)$ approximation achieved by variance penalization; the latter quantities for instance typically add an extra $\log(n)$ factor in the leading term in the case of VC classes and create even larger deteriorations as compared to the entropy integral for larger classes; achieving an entropy integral dependence has been an open question in the variance penalization literature and the f-divergence robust optimization formulations do not provide an answer to these as they similarly have dependence on metric entropy quantities at fixed approximation levels; moreover the f-divergence equivalence to variance penalization is only asymptotic, ii) we depend on the variance of the difference of the policy loss between the optimal policy and any policy in a small regret slice; this constant can be much smaller than the variance of the optimal policy that is achieved by the moment penalization of (FS) (see discussion after theorem 1).

*Computational efficiency.* We note that in both the examples that we present the policy learning problem is convex. We agree that we omitted the description of how we optimize the policy in the pricing example. But we do have a concrete discussion of how we optimize the policy in the costly resource allocation. In the case of pricing, where we optimize over linear policies, then observe that the problem is convex with respect to the coefficients in the linear policy (as it is of the form of maximizing: $\langle\gamma, z\rangle(a(z) + b(z)\langle\gamma, z\rangle)$ and $b(z)$ is non-positive; hence the hessian with respect to gamma is negative semi-definite and hence a concave maximization problem). In this case we optimized the objective by simply finding a closed form solution to the first order condition. This involves simple matrix computations. Similarly as we describe in the costly resource allocation application in Appendix G, the policy learning problem boils down to the square loss minimization over a space of high-dimensional linear policies subject to an $\ell_1$ ball constraint (e.g. a multi-task lasso problem); see Equation (55) and statement below. This is a convex problem and can be solved efficiently with standard packages; which is what we employed. In both cases, out-of-sample regularized ERM preserves the convexity of the ERM problem and is efficiently computable via convex optimization; as opposed to variance/moment penalization, which becomes a non-convex problem.

[Meta-Review · NeurIPS 2019]

All the reviewers enjoyed your paper and recommended acceptance. Congratulations! Please try to ensure that the clarifications promised in the rebuttal make their way into the camera ready.